# A theoretical framework for controlling complex microbial communities

Marco Tulio Angulo [1], Claude H. Moog [2] & Yang-Yu Liu [3,4]

Microbes form complex communities that perform critical roles for the integrity of their environment or the well-being of their hosts. Controlling these microbial communities can help us restore natural ecosystems and maintain healthy human microbiota. However, the lack of an efficient and systematic control framework has limited our ability to manipulate these microbial communities. Here we fill this gap by developing a control framework based on the new notion of structural accessibility. Our framework uses the ecological network of the community to identify minimum sets of its driver species, manipulation of which allows controlling the whole community. We numerically validate our control framework on large communities, and then we demonstrate its application for controlling the gut microbiota of gnotobiotic mice infected with *Clostridium difficile* and the core microbiota of the sea sponge *Ircinia oros*. Our results provide a systematic pipeline to efficiently drive complex microbial communities towards desired states.

[1] CONACyT – Institute of Mathematics, Universidad Nacional Autónoma de México, Juriquilla, Querétaro 76230, Mexico. [2] Laboratoire des Sciences du Numérique de Nantes, UMR CNRS 6004, Nantes 44321, France. [3] Channing Division of Network Medicine, Brigham and Women's Hospital and Harvard Medical School, Boston, MA 02115, USA. [4] Center for Cancer Systems Biology, Dana-Farber Cancer Institute, Boston, MA 02115, USA. Correspondence and requests for materials should be addressed to M.T.A. (email: mangulo@im.unam.mx) or to Y.-Y.L. (email: yyl@channing.harvard.edu)

Microorganisms form complex communities that play critical roles in maintaining the well-being of their hosts or the integrity of their environment[1–4]. Disrupting these microbial communities can have severe consequences. In humans, for example, a disruption to the gut microbiota—the aggregate of microorganisms residing in our intestine—is associated with several disorders including irritable bowel syndrome, *Clostridium difficile* Infection (CDI), autism, obesity, and cavernous cerebral malformations[5–7]. For agriculture crops, a disruption of rhizosphere microbiota can reduce their disease resistance and hence decrease the overall crop yield[8,9]. In the oceans, a disruption to their microbiota can impact global climate by altering carbon sequestration rates[3,4,10]. Driving disrupted microbial communities back to their healthy states could offer novel solutions to prevent and treat complex human diseases, enhance sustainable agriculture, and regulate global warming[11,12]. For instance, inoculating soil microbes can restore terrestrial ecosystems[13], and fecal microbiota transplantation (FMT) is so far the most successful therapy for treating recurrent CDI[14]. Despite the success of these empirical strategies, a broad application of microbial-manipulation strategies will be possible only if we can efficiently control large complex microbial communities[15].

There are two big challenges down the road. First, an efficient control method should only manipulate a minimum set of species in the community. However, we still lack a systematic method to identify minimum sets of those "driver species" whose control can help us drive a whole community to desired states. Here, we use the term "species" without necessarily representing the lowest major taxonomic rank. One could also organize microbes by strains, genera, or operational taxonomical units. Second, even when those driver species have been identified, designing the control strategy that should be applied to them (e.g., how their abundance needs to be manipulated) for driving the community towards the desired state remains difficult. This difficulty arises because of the inherent complexity of microbial dynamics and our limited knowledge of them.

To address those two challenges, here we develop a control framework using the ecological network underlying the microbial community. First, we introduce the new notion of "structural accessibility", which generalizes the notion of structural linear controllability[16,17] to systems with nonlinear dynamics. Then, we derive a complete graph-theoretical characterization of structural accessibility. This result enables us to efficiently identify minimum sets of driver species of any microbial community purely from the topology of its underlying ecological network, even if some microbial interactions are missing and its population dynamics is unknown. Once the driver species are identified, we systematically design feedback control strategies to drive a microbial community towards the desired state, even if its dynamics is not precisely known. We numerically validated our control framework in large microbial communities, analyzing its performance for different parameters of the community (e.g., the connectivity of its underlying ecological network), and for errors in the ecological network used to identify the driver species. Finally, we demonstrate our framework by controlling the core microbiota of the sea sponge *Ircinia oros*, and restoring the gut microbiota of gnotobiotic mice infected by *Clostridium difficile*. Our results provide a rational and systematic framework to control microbial communities and other complex ecosystems.

## Results
### Modeling controlled microbial communities
Our framework focuses on the impact that manipulating a subset of species has on the abundances of other species. We thus consider a microbial community whose state at time $t$ is determined from the abundance profile $x(t) \in \mathbb{R}^N$ of its $N$ species, where the $i$-th entry $x_i(t)$ represents the abundance of the $i$-th species at time $t$. The state evolves according to some population dynamics

$$\dot{x}(t) = f(x(t)), \qquad (1)$$

where the function $f : \mathbb{R}^N \rightarrow \mathbb{R}^N$ models the species intrinsic growth and the inter/intra-species interactions of the community (see Supplementary Note 1 for details). For most microbial communities $f$ is unknown and difficult to infer given the many interaction mechanisms between microbes[18]. Thus, we assume that $f(x)$ is some unknown meromorphic function of $x$ (i.e., the quotient of analytic functions). This assumption is very mild as it is satisfied by most population dynamics models[19].

Instead of knowing the population dynamics of the microbial community, we assume we know its underlying ecological network $\mathcal{G} = (X, E)$. This network is a directed graph where nodes $X = \{x_1, \ldots, x_N\}$ represent species, and edges $(x_j \rightarrow x_i) \in E$ denote that the $j$-th species has a direct ecological impact (i.e., direct promotion or inhibition of growth) on the $i$-th species (Fig. 1a). Mapping ecological networks requires performing mono- and co-culture experiments[20,21], using system identification techniques with time-resolved abundance data[22,23], or using steady-state abundance data via a recently developed inference method[24]. In general, ecological networks are different from correlation networks[20,25] because correlation does not imply causation[26,27].

Controlling the community consists in driving its state from an initial value $x_0 = x(0) \in \mathbb{R}^N$ at $t = 0$ (e.g., a "diseased" state) towards the desired value $x_d \in \mathbb{R}^N$ (e.g., the "healthy" state, Fig. 1b). We assume that the community will not evolve by itself to $x_d$. To drive the community, we use $M$ control inputs $u(t) \in \mathbb{R}^M$ directly affecting certain species that we call "actuated species" (Fig. 1a). Control inputs encode a combination of $M$ control actions applied at time $t$. We consider four possible control actions. If $u_j(t) < 0$, the $j$-th control action at time $t$ can be a bacteriostatic agent or bactericide, decreasing the abundance[28] of the species it actuates. If $u_j(t) > 0$, the $j$-th control action at time $t$ can be a prebiotic[29] or transplantation, stimulating the growth or engrafting a consortium of the species it actuates, respectively. Probiotics administration[30] and FMTs[14] are examples of transplantations. To specify the species actuated by each control input we introduce the controlled ecological network $\mathcal{G}^c = (X \cup U, E \cup B)$. Here, $U = \{u_1, \ldots, u_M\}$ are the control input nodes and $(u_j \rightarrow x_i) \in B$ denotes that the $j$-th control input actuates the $i$-th species (Fig. 1a).

We introduce two control schemes describing how the control inputs change the species abundance (see Supplementary Note 1 for details). The first control scheme models a combination of prebiotics (if $u_j(t) > 0$) and bacteriostatic agents (if $u_j(t) < 0$) as continuous control inputs modifying the growth of the actuated species (Fig. 1c):

$$\dot{x}(t) = f(x(t)) + g(x(t))u(t), \quad t \in \mathbb{R}. \qquad (2)$$

The second control scheme models a combination of transplantations (if $u_j(t) > 0$) and bactericides (if $u_j(t) < 0$) applied at discrete intervention instants $\mathbb{T} = \{t_1, t_2, \cdots\}$, rendering impulsive control inputs that instantaneously modify the

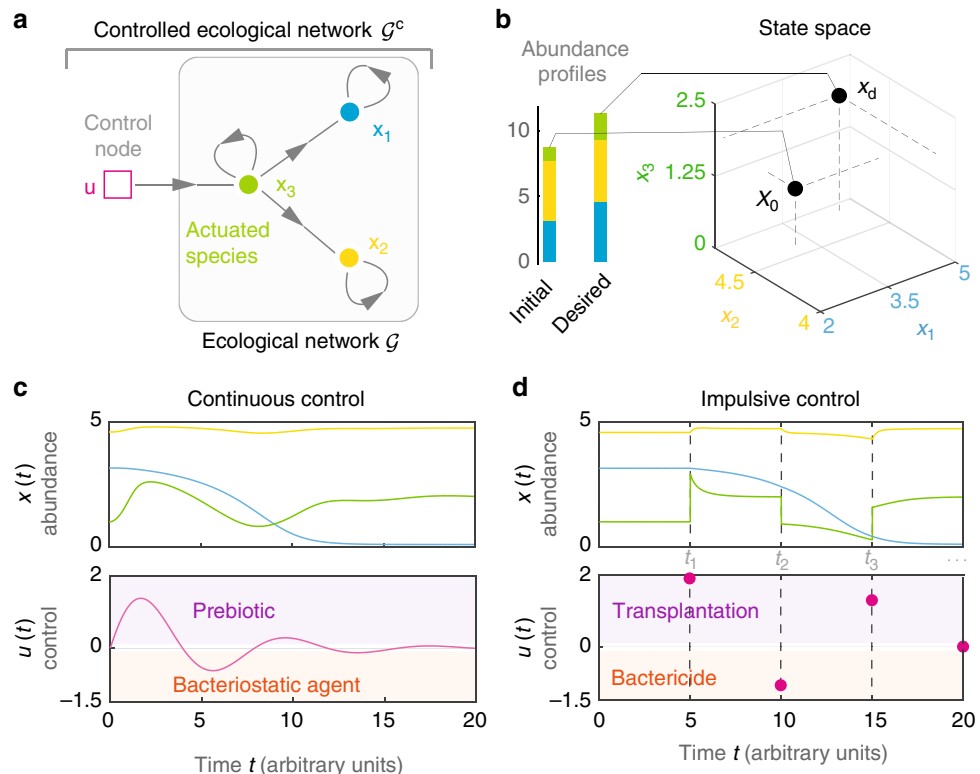

**Fig. 1** Controlling a microbial community. **a** Ecological network $\mathcal{G}$ for a toy microbial community of $N = 3$ species (green, yellow, blue). The controlled ecological network $\mathcal{G}^c$ contains $M = 1$ control input actuating the third species. **b** Initial and desired abundance profiles (bars). Controlling the community consists in driving its state from the initial state $x_0$ to the desired state $x_d$, represented by two points in the state space of the community. **c** In the continuous control scheme, the control inputs $u(t)$ are continuous signals modifying the growth of the actuated species. The controlled population dynamics of this community is given by $\dot{x}_1 = 0.1 + x_1(1 - x_1/5)(x_1/3 - 1) - (0.1x_1x_3)/(1 + x_3)$, $\dot{x}_2 = 0.1 + x_2(1 - x_2/4)(x_2 - 1) + (x_2x_3)/(1 + x_3)$, $\dot{x}_3 = x_3(1 - x_3/2)(x_3 - 1) + u$. In the absence of control, this community has two equilibria $x_0 = (3.14, 4.58, 1)^\top$ and $x_d = (4.57, 4.73, 2)^\top$, chosen as the initial and desired states, respectively. **d** In the impulsive control scheme, the control inputs $u(t)$ are impulses applied at the intervention instants $\mathbb{T} = \{t_1, t_2, \cdots\}$, instantaneously changing the abundance of the actuated species. The controlled population dynamics is the same as in panel (**c**), except that $\dot{x}_3 = x_3(1 - x_3/2)(x_3 - 1)$ and $x_3(t^+) = x_3(t) + u(t)$ if $t \in \mathbb{T} = \{5, 10, 15\}$. Under this controlled population dynamics, our mathematical formalism identifies $x_3$ as the solo driver species needed to drive this microbial community (Example 1 in Supplementary Note 2)

abundance of the actuated species (Fig. 1d):

$$\dot{x}(t) = f(x(t)) \text{ if } t \notin \mathbb{T}, \quad x(t^+) = x(t) + g(x(t))u(t) \text{ if } t \in \mathbb{T}. \tag{3}$$

Above, $x(t^+)$ denotes the state "right after time $t$", so $x(t)$ "jumps" at $t \in \mathbb{T}$ if $u(t) \neq 0$. The pair $\{f, g\}$ characterizes both control schemes, describing the controlled population dynamics of the microbial community. The function $g : \mathbb{R}^N \to \mathbb{R}^{N \times M}$ models the direct susceptibility of the species to the control actions. The $j$-th control input actuates the $i$-th species if $g_{ij} \not\equiv 0$. Because $g$ is typically unknown, we just assume that $g(x)$ is some unknown meromorphic function of $x$ such that $g_{ij} \not\equiv 0$ iff $(u_j \to x_i) \in B$.

Notice that when all species are directly controlled (i.e., an independent control input actuates each species), the whole microbial community can easily be driven to the desired state. Fortunately, as we show next, actuating all the species is far from being necessary. Thanks to the inter-species interactions encoded in the ecological network $\mathcal{G}$, we can identify minimum sets of species that we need to actuate in order to drive the whole community. We call those species "driver species".

**Identifying driver species.** To understand when a set of actuated species is a set of driver species, consider the three-species community with Generalized Lotka–Volterra (GLV) population dynamics of Fig. 2a. This toy community has one control input actuating $x_3$. Actuating only this species creates an autonomous element—namely, a constraint between some species abundances that the control input cannot break, confining the state of the community to a low-dimensional manifold (Fig. 2a, right). More precisely, our mathematical formalism reveals that $\xi = x_1x_2$ is the autonomous element (Example 2 in Supplementary Note 2). Indeed, differentiating $\xi$ with respect to time yields $\dot{\xi} = x_1x_2(1 - x_3) + x_1x_2(-1 + x_3) \equiv 0$, confining the community to $\{x \in \mathbb{R}^3 | x_1x_2 = x_1(0)x_2(0)\}$. Intuitively, the autonomous element exists because the control input cannot change $x_1$ without changing $x_2$ in a predefined way, making it impossible to drive the community in the three-dimensional state space. This observation indicates that $x_3$ alone cannot be a driver species for this community. Introducing a second control input actuating $x_1$ helps the community jump out of the low-dimensional manifold eliminating the autonomous element, allowing us to drive this community to any desired state with positive abundance (Fig. 2b, and Example 6 in Supplementary Note 5). Therefore, $\{x_1, x_3\}$ is a minimum set of driver species for this community.

In the general case of $N$ species and $M$ control inputs, we define a set of actuated species as a set of driver species if the corresponding controlled population dynamics $\{f, g\}$ lacks autonomous elements. For linear dynamics $\{f(x), g(x)\} = \{Ax, B\}$, $A \in \mathbb{R}^{N \times N}$, $B \in \mathbb{R}^{N \times M}$,

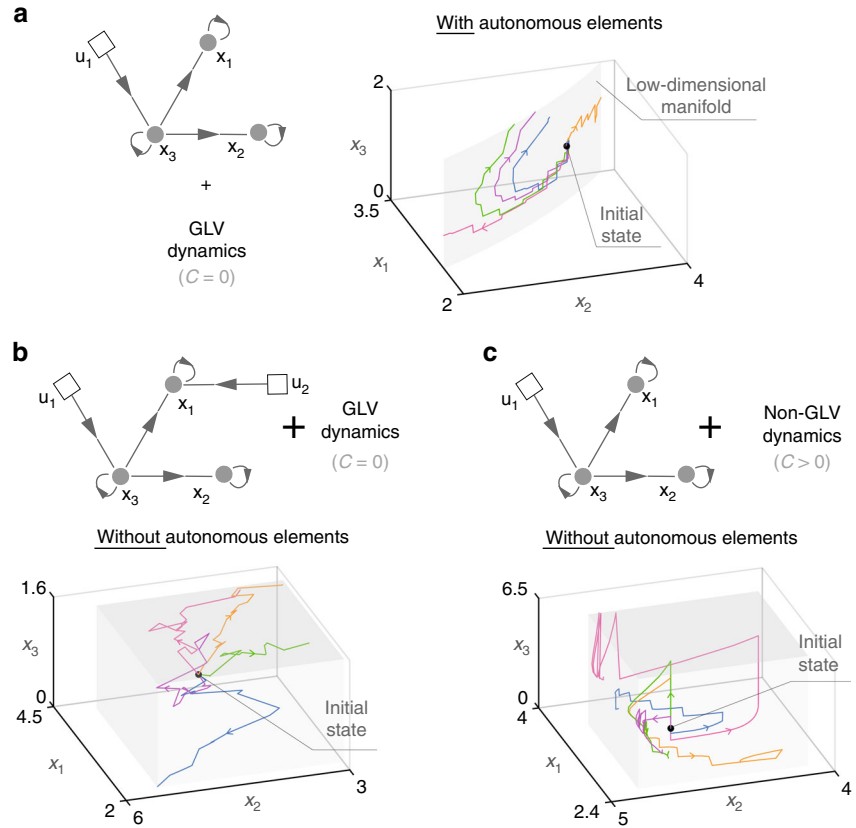

**Fig. 2** Autonomous elements constrain the state of microbial communities, characterizing their driver species. **a** A three-species community with GLV dynamics $\dot{x}_1 = x_1(-1 + x_3)$, $\dot{x}_2 = x_2(1 - x_3)$, $\dot{x}_3 = x_3(-0.5 + 1.5x_3)$. For actuating $x_3$, we consider the impulsive control scheme with $x_3(t^+) = x_3(t) + u_1(t)$ for $t \in \mathbb{T}$. With this controlled population dynamics, our mathematical formalism reveals the autonomous element $x_1 x_2$ that constraints the state of this microbial community to the low-dimensional manifold $\{x \in \mathbb{R}^3 | x_1 x_2 = x_1(0)x_2(0)\}$ (gray) for all control inputs. Five state trajectories (in colors) with random control inputs illustrate this fact. Hence, $\{x_3\}$ alone cannot be a set of driver species for this controlled population dynamics. **b** Including a second control input $u_2(t)$ actuating $x_1$ (i.e., $x_1(t^+) = x_1(t) + u_2(t)$ for $t \in \mathbb{T}$) eliminates the autonomous element, since the state of the microbial community (colors) can explore a three-dimensional space (gray). Hence $\{x_1, x_3\}$ is a minimum set of driver species for this community with GLV dynamics. **c** We proved that, generically, increasing the complexity of the controlled population dynamics cannot create autonomous elements. In this example, increasing the deformation size $C$ from the GLV in panel (**a**) (with $C = 0$) to the controlled population dynamics in Fig. 1 (with $C > 0$) eliminates the autonomous element that was present by actuating $x_3$ alone (Example 1 in Supplementary Note 2). Therefore, increasing the complexity of the population dynamics makes $\{x_3\}$ a solo driver species

the absence of autonomous elements is equivalent to their controllability[31]—the ability to drive the system between any two states, easily verified using Kalman's condition rank $[B, AB, \cdots, A^{N-1}B] = N$. In the case of nonlinear dynamics, the absence of autonomous elements can be characterized using a mathematical formalism based on differential one-forms (see Methods and Supplementary Note 2). For the continuous control scheme of Eq. (2), the conditions for the absence of autonomous elements are well understood as they define when a system is accessible[31], a cornerstone concept in nonlinear control theory. Because it is more natural to control microbial communities with impulsive control actions, in this paper we extended the study of autonomous elements to the impulsive control systems of Eq. (3). We first introduced a definition of autonomous elements for impulsive control systems (Definition 3 in Supplementary Note 2). We then characterized necessary and sufficient conditions for the absence of autonomous elements in a controlled population dynamics (Theorem 2 in Supplementary Note 2). To our surprise, the conditions for the absence of autonomous elements for the continuous and the impulsive control schemes are identical (Remark 2 in Supplementary Note 2). This result means that transplantations and bactericides (impulsive control actions) can

be as effective as prebiotics and bacteriostatic agents (continuous control actions).

**Structural accessibility characterizes the generic absence of autonomous elements.** In general, it remains extremely difficult finding a pair $\{f, g\}$ that models the controlled population dynamics of a microbial community. This fact might suggest it is impossible to predict if the controlled community has autonomous elements or not, making it impossible to identify its driver species. We now show that this seemingly unavoidable limitation can be solved using the topology of the controlled ecological network of the community.

Define the network $\mathcal{G}_{f,g} = (X \cup U, E_{f,g} \cup B_{f,g})$ associated with $\{f, g\}$ as follows: $(x_j \rightarrow x_i) \in E_{f,g}$ if $x_j$ appears in the right-hand side of $\dot{x}_i$ in Eq. (2) or $x_i(t^+)$ in Eq. (3). Similarly, $(u_j \rightarrow x_i) \in B_{f,g}$ if $g_{ij} \not\equiv 0$. Using this definition, we next describe the class $\mathfrak{D}$ of all possible controlled population dynamics that a controlled microbial community can have given we know its $\mathcal{G}^c$. Mathematically, $\mathfrak{D}$ contains all base models $\{f^*, g^*\}$ such that $\mathcal{G}_{f^*,g^*} = \mathcal{G}^c$, together with all deformations $\{f, g\}$ of each of those base models. The base models characterize the simplest controlled population dynamics that the community can have, leading us to choose

them as controlled GLV models with constant susceptibilities:

$$f_i^*(x) = r_i x_i + \sum_{j=1}^{N} a_{ij} x_i x_j, \quad g_{ij}^*(x) = b_{ij}, \tag{4}$$

for $i = 1, \dots, N$. The parameters $A = (a_{ij}) \in \mathbb{R}^{N \times N}$, $r = (r_i) \in \mathbb{R}^N$, and $B = (b_{ij}) \in \mathbb{R}^{N \times M}$ represent the interaction matrix, the intrinsic growth rate vector, and the susceptibility matrix of the community, respectively. As the simplest population dynamics, the GLV model has been applied to microbial communities in lakes, soils, and human bodies[14,15,20,32–38].

A deformation of $\{f^*, g^*\}$ is any meromorphic pair $\{f, g\}$ such that: (i) $\mathcal{G}_{f,g} = \mathcal{G}_{f^*,g^*}$; (ii) there exists a finite set of parameters $\theta \in \mathbb{R}^C$ such that $\{f(x), g(x)\} = \{\tilde{f}(x;\theta), \tilde{g}(x;\theta)\}$; and (iii) the identity $\{\tilde{f}(x;0), \tilde{g}(x;0)\} = \{f^*(x), g^*(x)\}$ holds. The smallest integer $C \geq 0$ satisfying these three conditions is called the size of the deformation. A general class of controlled population dynamics are deformations of Eq. (4), including

$$f_i(x;\theta) = \theta_{i,1} + x_i \left( -r_i - \theta_{i,2} x_i \right) \left( \theta_{i,3} x_i - 1 \right)$$
$$+ \sum_{j=1}^{N} a_{ij} \frac{x_i x_j}{1 + \theta_{ij,4} + \theta_{ij,5} x_i + \theta_{ij,6} x_i x_j + \theta_{ij,7} x_j}, \tag{5}$$

for $i = 1, \dots, N$. Above, $\theta_{i,1}$ are migration rates from/to neighboring habitats, $\theta_{i,2}^{-1}$ are the carrying capacities of the environment, $\theta_{i,3}^{-1}$ are the Allee constants, and $\{\theta_{ij,k}\}_{k=4}^{7}$ characterize the functional responses[39]. $\theta_{i,1} > 0$ also models species like *C. difficile* that sporulate into "inactive" forms and then recover. "Higher-order interactions" (e.g., $\theta_i x_i x_j x_k$) and susceptibilities mediated by species abundance (e.g., $g_{ij}(x;\theta) = b_{ij} + \theta_{ijk} x_k$) are deformations as well.

We call $\mathfrak{D}$ structurally accessible if almost all of its base models and almost all of their deformations lack autonomous elements. This definition means that except for a zero-measure set of "singularities," all the controlled population dynamics that the community may take have to lack autonomous elements. The conditions under which $\mathfrak{D}$ is structurally accessible are fully characterized using our mathematical formalism and they depend only on $\mathcal{G}^c$ (see Methods and Supplementary Note 3). Hence, if $\mathfrak{D}$ is structurally accessible, hereafter we also call $\mathcal{G}^c$ structurally accessible. We first proved that, generically, increasing the size of a deformation cannot create autonomous elements (Proposition 1 in Supplementary Note 3). See also Fig. 2c for an illustration. This result reduces the search for autonomous elements to the deformations in $\mathfrak{D}$ with minimum size $C = 0$ (i.e., all base models whose graph matches $\mathcal{G}^c$). Finally, we proved that $\mathfrak{D}$ is structurally accessible if and only if $\mathcal{G}^c$ satisfies the following two graph–theoretical conditions: (i) each species is the end-node of a path that starts at a control input node; and (ii) there is a disjoint union of cycles (excluding self-loops) and paths that cover all species nodes (Theorem 3 of Supplementary Note 3). Note that the conditions for structural accessibility depend on the chosen base model.

Structural accessibility is a nonlinear generalization of "structural controllability" for linear systems[16]. The latter notion has received increasing attention in Network Science[17]. Interestingly, the two graph–theoretical conditions for structural accessibility are almost the same as those for structural linear controllability[16]. The key difference is that for structural linear controllability self-loops (corresponding to intrinsic nodal dynamics) can be used to satisfy condition (ii). See Remark 4 in Supplementary Note 3 for more details.

**Identifying minimum sets of driver species in microbial communities**. The above result provides a complete graph-characterization of driver species: a set of actuated species is a set of driver species (for all but a zero-measure set of controlled population dynamics that the community may have) if and only if its corresponding $\mathcal{G}^c$ satisfies the two graph–theoretical conditions. See Fig. 3 for an illustration. With this characterization, one can apply the maximum matching algorithm directly to $\mathcal{G}$ to calculate the minimum number of control inputs needed to ensure the structural accessibility of $\mathcal{G}^c$, as did in the structural linear controllability case[17,40]. However, this may not provide a minimum set of driver species because one control input may actuate multiple species. Fortunately, we can dedicate one control input to one species. Therefore, we adapted the notion of a

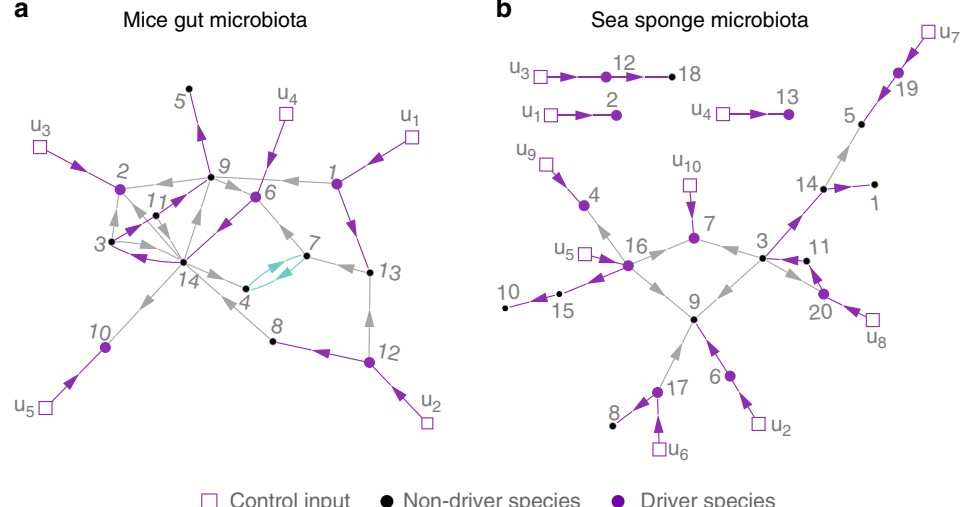

**Fig. 3** Identifying driver species. For each network, a minimum set of driver species is shown providing a disjoint union of paths (purple) and cycles (green) covering all species nodes (see Supplementary Table 1 for the species name). Thus, the resulting controlled ecological network is structurally accessible. Self-loops are omitted from these networks to improve readability. **a** Inferred ecological network of the gut microbiota of germ-free mice pre-colonized with a mixture of human commensal bacterial type strains and then infected with *C. difficile* (species 7), as in ref. [22] **b** Inferred ecological network of the core microbiota of the sea sponge *Ircinia oros*, as in ref. [23]

"feasible dedicated input configuration"[41] and a polynomial-time algorithm (combining maximum matching with a strongly connected component decomposition of $\mathcal{G}$) to identify one minimum set of driver species (Methods and Supplementary Note 4). Note that once $\mathcal{G}^c$ is structurally accessible, it cannot lose its structural accessibility when new edges are added to it. This observation implies that the driver species can be identified from an "incomplete" ecological network (e.g., containing only high-confidence interactions).

**Driving the driver species.** We next calculate the control inputs to be applied to a set of driver species for driving the whole community towards the desired state $x_d$. We will show that it is more efficient to calculate impulsive control inputs. To calculate these impulsive control inputs $\{u(t_k), t_k \in \mathbb{T}\}$ we adopt a model predictive control (MPC) approach[42]. Based on the current state of the community $x(t_k)$ at $t_k \in \mathbb{T}$, we use knowledge of its controlled population dynamics $\{f, g\}$ to predict the sequence of states $\hat{X}_{k,L} = \{\hat{x}(t_{k+1}), \cdots, \hat{x}(t_{k+L+1})\}$ that the community will take in response to a sequence of $L$ impulsive control inputs $U_{k,L} = \{u(t_k), \cdots, u(t_{k+L-1})\}$. The prediction horizon $L > 0$ determines how far into the future we predict. Then, we choose $u(t_k) = u_1^*(t_k)$ where $u_1^*(t_k)$ is the first element of the optimal control sequence $U_{k,L}^*$ calculated as:

$$U_{k,L}^* = \arg \min_{U_{k,L} \in \mathbb{R}^{M \times L}} J_{x_d}(\hat{X}_{k,L}, U_{k,L}) \text{ subject to } U_{k,L} \in \Omega. \quad (6)$$

Here, $\Omega \subseteq \mathbb{R}^{M \times L}$ specifies constraints in the control inputs, and $J_{x_d}$ is some cost function penalizing deviations of the predicted trajectory $\hat{X}_{k,L}$ from $x_d$. For example, the cost function $J_{x_d}(\hat{X}_{k,L}, U_{k,L}) = \|\hat{x}(t_{k+L+1}) - x_d\|$ penalizes the deviations of the predicted final state. By recalculating $U_{k,L}^*$ at each $t_k$ using the actual state of the community the MPC creates a feedback loop enhancing its robustness against prediction errors[42]. The prediction horizon can be chosen based on the controlled population dynamics of the community (Methods). For $L = 1$, this methodology is similar to ref. [43]. Equation (6) is a finite-dimensional optimization problem that can be solved using algorithms like DIRECT[44]. Solving the analogous optimization problem for continuous control inputs is more challenging because the optimization is over the infinite-dimensional space of continuous functions.

We illustrate the above MPC strategy driving the microbial community of Fig. 1 with its solo driver species. According to its dynamics, $L = 3$ impulsive control inputs are sufficient (see caption in Fig. 1, and Example 4 in Supplementary Note 5). We chose $J_{x_d}(\hat{X}_{k,L}, U_{k,L}) = \|\hat{x}(t_{k,L}) - x_d\|_2$. Solving Eq. (6) using DIRECT yields the nonlinear MPC strategy $u^*(t_1) = -0.8815$, $u^*(t_2) = 2.0089$ and $u^*(t_3) = -10^{-4}$ (pink in Fig. 4a). We compared the performance of two other control strategies. The first strategy uses one transplantation to increase the abundance of the driver species to its desired value, reminiscent of one probiotic administration restoring its "healthy" abundance (purple in Fig. 4a). The second control strategy ignores the driver species, setting the abundance of the two non-driver species to their desired values (blue in Fig. 4a).

Among the above three control strategies, only the nonlinear MPC applied to the driver species succeeds (Fig. 4b). This strategy succeeds in a somewhat unconventional way: although the driver species is more abundant in the desired state than in the initial state, the first control action decreases its abundance further. Such control action lets the non-driver species reach their desired abundances and, once that happens, the abundance of the driver species is finally increased to its desired value (pink in Fig. 4b). Just restoring the abundance of the driver species succeeds in

driving $x_2$ and $x_3$, but it fails to drive $x_1$ to the desired abundance (purple in Fig. 4b). Ignoring the driver species is the worst control strategy, failing to drive any of the three species to their desired values (blue in Fig. 4b). This toy example demonstrates the advantage of identifying and actuating driver species.

**Driving large communities with uncertain dynamics.** Solving the non-convex optimization problem of Eq. (6) is challenging as $N$ or $L$ increase, and it also requires knowing $\{f, g\}$, which may be impossible for large communities. We next circumvent these two drawbacks leveraging the network underlying the controlled microbial community.

Consider we can obtain a weighted adjacency matrix $\hat{A} \in \mathbb{R}^{N \times N}$ from $\mathcal{G}$, providing a proxy for its interaction matrix. Without additional knowledge of the community, we just assume that we can increase or decrease the abundance of each driver species. We thus use $\hat{B} \in \{0, 1\}^{N \times M}$ as a proxy for the susceptibility matrix, with $b_{ij} = 1$ if the $j$-th control input actuates the $i$-th driver species. By rewriting $\{f(x), g(x)\} = \{\hat{A}x + w_x, \hat{B} + w_u\}$, we use $\{\hat{A}x, \hat{B}\}$ to provide a linear prediction for the response of the community to the control inputs. Here, $w_x = f - \hat{A}x$ and $w_u = g - \hat{B}$ are considered as "perturbations". Using $\{\hat{A}x, \hat{B}\}$, we design a linear MPC by solving Eq. (6) with the quadratic cost function

$$J_{x_d}(\hat{X}_{k,\infty}, U_{k,\infty}) = \sum_{i=k}^{\infty} [\hat{x}(t_i) - x_d]^\top Q[\hat{x}(t_i) - x_d] + u(t_i)^\top R u(t_i).$$

Above, the positive definite matrices $Q = Q^\top \in \mathbb{R}^{N \times N}$ and $R = R^\top \in \mathbb{R}^{M \times M}$ are design parameters. $Q$ penalizes the deviations of the predicted trajectory from the desired state, and $R$ penalizes the control inputs magnitude. Under this scenario, Eq. (6) can be solved in closed form[45] yielding the linear MPC $u(t_k) = K x(t_k)$, where $K \in \mathbb{R}^{M \times N}$ is the solution of a Riccati equation (Supplementary Note 6). Since the Ricatti equation can be efficiently solved for large $N$, the linear MPC can be calculated for large communities. This linear MPC is robust against $(w_x, w_u)$ and it allows calculating the control inputs for the continuous control scheme (Supplementary Note 6). However, its performance strongly depends on the chosen $(\hat{A}, \hat{B})$ and the distance to the desired state (Supplementary Note 6).

We applied the linear MPC for driving the toy three-species community of Fig. 1, assuming its dynamics is uncertain. Considering the ecological network of this community and its nonlinear population dynamics, we chose $\hat{A} = (-0.5, 0, -0.1; 0, -5, 1; 0, 0, -1)$ as a proxy for its interaction matrix. Here $\hat{A}$ is a rough approximation of the linearization of the population dynamics at the desired state given by $(-0.37, 0, -0.05; 0, -5.31, 0.52; 0, 0, -1)$. Choosing $Q = \text{diag}(20, 1, 10)$, we compared the performance of three different linear MPCs obtained with $R = 10^{-4}, 10^{-3}, 10^{-2}$ (Fig. 4c). For $R = 10^{-4}$, without using knowledge of the population dynamics, the performance of the linear MPC (pink in Fig. 4d) is very similar to the performance of the nonlinear MPC that uses full knowledge of the nonlinear population dynamics (pink in Fig. 4b). This success illustrates the robustness of the linear MPC against the perturbations. As $R$ increases, the performance of the linear MPC deteriorates (green and blue in Fig. 4d).

**Numerical validation on large uncertain microbial communities.** To validate our control framework for large communities, we built communities of $N = 100$ species having random directed Erdös–Rényi ecological networks with connectivity $c \in [0, 1]$, see

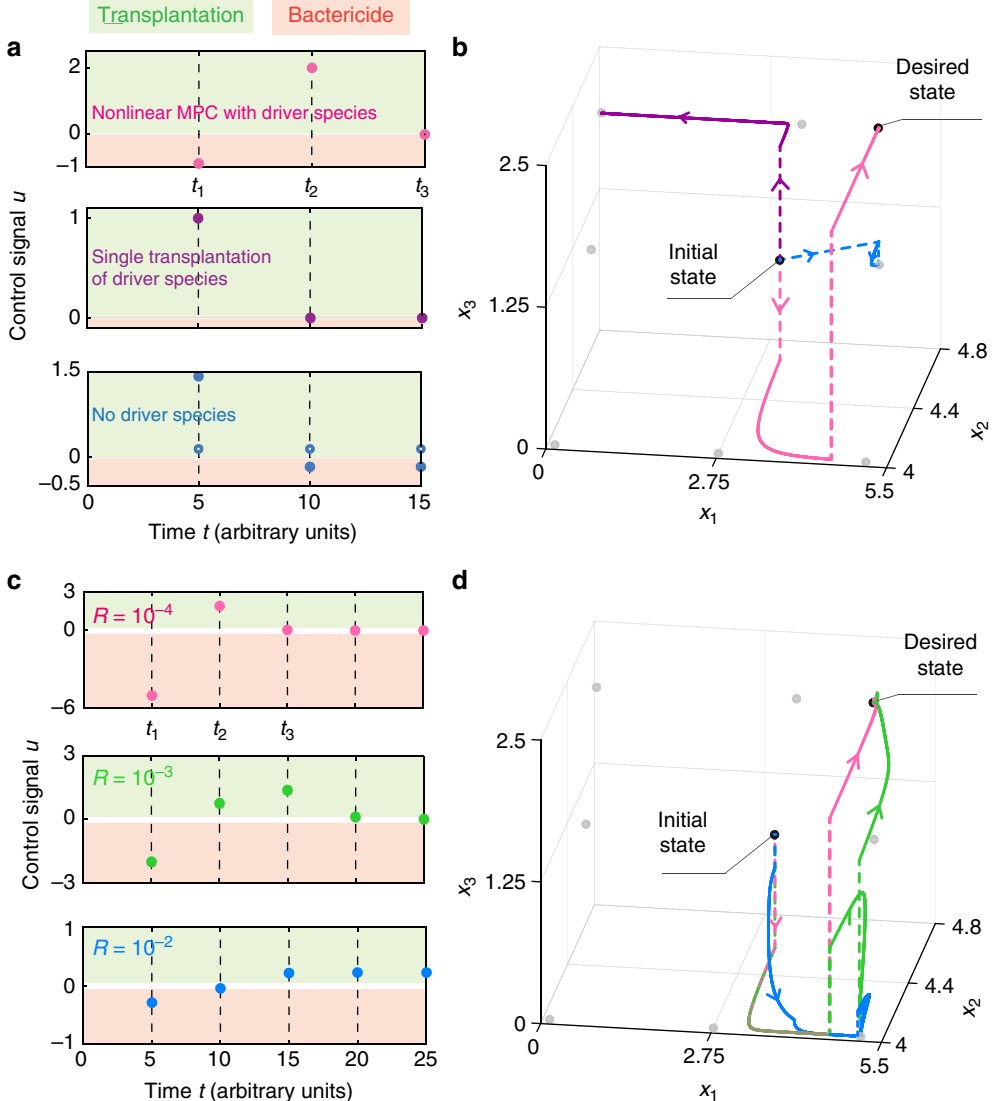

**Fig. 4** Success and failure of different control strategies. **a** Three control strategies for driving the microbial community of Fig. 1a toward the desired state. First, MPC applied to the identified driver species $\{x_3\}$ (pink dots). The second control strategy increases the abundance of the driver species to match its value at the desired state $x_3(t_1) = x_{3,d}$ (purple dots). The third control strategy does not actuate the driver species, but actuates the other two species $\{x_1, x_2\}$ by setting their abundance to their desired values (i.e., $x_1(t_k) = x_{1,d}$ and $x_2(t_k) = x_{2,d}$, solid and hollow blue dots, respectively). **b** Response of the microbial community to these three control strategies. Here and in panel (**d**), the "jumps" produced by the control inputs are depicted by dashed lines. The equilibria of the population dynamics are shown as gray dots. Only the MPC applied to the driver species succeeds in driving the community to $x_d$. **c** Control strategies obtained by using the linear MPC with parameters $Q = \mathrm{diag}(20, 1, 10)$ and different values for $R$: $10^{-4}$ (pink), $10^{-3}$ (green), $10^{-2}$ (blue). **d** Trajectories of the controlled community using the linear MPC strategies described in panel (**c**). Colors correspond to the different values of $R$

Fig. 5a. The network edge-weights were chosen from a normal distribution with zero mean and standard deviation $\sigma > 0$, where $\sigma$ characterizes the typical interspecies interaction strength. Negative self-loops with weights $-1$ were added to each species. We used this ecological network to identify the driver species of the community, and its corresponding weighted adjacency matrix as the interaction matrix to construct the linear MPC. We simulated the population dynamics of these communities using Eq. (5) ensuring all share $x_d \in \mathbb{R}^N$ as equilibrium. The resulting communities have nonlinear population dynamics, and their linearization at the desired state is different from the interaction matrix used for the linear MPC (Supplementary Note 8).

To quantify the success of our control framework on a given community, we generated 300 initial species abundances that are uniformly distributed at a distance $d > 0$ from $x_d$. The success rate at distance $d$ is defined as the proportion of those initial

conditions that are driven to $x_d$ only when the linear MPC is applied to a minimum set of driver species of the community (Fig. 5b–d). Namely, we discard all initial conditions that naturally evolve to $x_d$. Finally, we calculated the mean success rate by averaging the success rate over 100 random communities (see Supplementary Note 8 for details).

The mean success rate is close to 1 for small $d$ regardless of the community's parameters (Fig. 5e, f), confirming the theoretical guarantee that the linear MPC succeeds if $d$ is small enough. The mean success rate decreases as $\sigma$ increases, especially for large distances (Fig. 5e). Since increasing $\sigma$ damages the stability of the population dynamics[46], this result suggests that microbial communities become "harder" to control as they lose stability. The mean success rate is higher in communities with low connectivity (Fig. 5f). In general, the size of a minimum set of driver species increases as $c$ decreases, indicating that the success

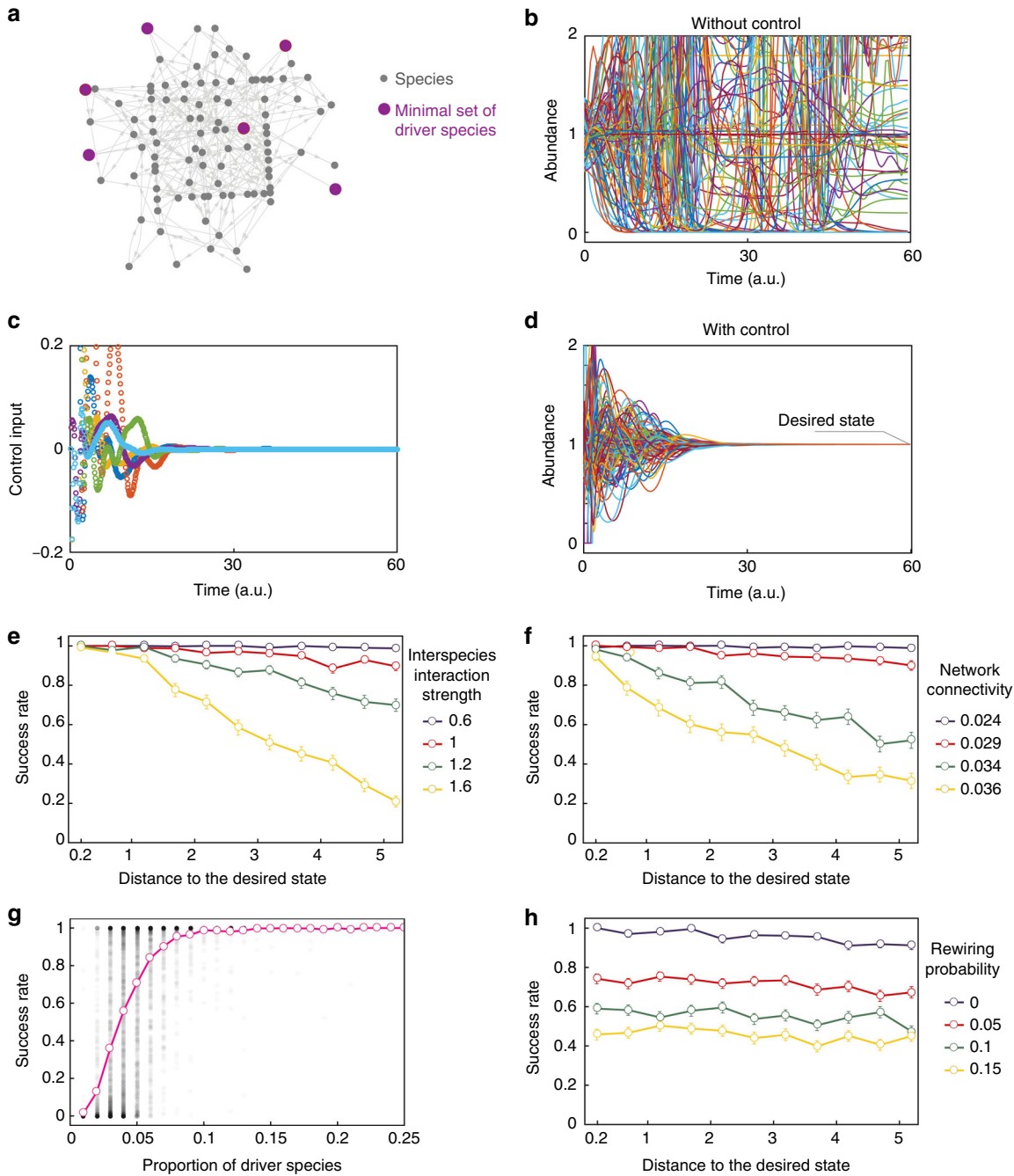

**Fig. 5** Numerical validation on large microbial communities. **a** Example of the ecological network for a random microbial community with $N = 100$ species ($c = 0.03$). A minimum set of $M = 6$ driver species is shown in purple. The desired state is chosen as $x_d = (1, \cdots, 1)^\top$. **b** With a random initial abundance $x_0$ at distance $d = 0.4$ from the desired state, the uncontrolled microbial community does not reach $x_d$. **c**, **d** For the same community and initial abundance as in panel (**b**), we apply the control input generated by the linear MPC (panel **c**) to the six identified driver species. This control strategy drives the community to $x_d$ (panel **d**). **e**, **f**, **h** Mean success rate as a function of $d$. Error bars denote the standard error of the mean. Parameters are: $c = 0.025$, $\theta_{max} = 0.05$ for panel (**e**), $\sigma = 0.8$, $\theta_{max} = 0.05$ for panel (**f**), and $c = 0.025$, $\sigma = 0.8$, $\theta_{max} = 0.05$ for panel (**h**). **g** Success rate as function of the proportion $M/N$ of driver species. Black dots show the success rate of 7700 random communities. Pink shows the mean success rate

rate increases as the number of driver species increases. Indeed, regardless of $d$, our control framework attains a mean success rate >0.8 provided that at least 6 from 100 species are driver species (Fig. 5g). This result suggests that the success rate can be enhanced by actuating a few additional species. Finally, to investigate the robustness of our control framework to errors in the ecological network, we randomly rewired each of its edges with probability $p \in [0,1]$ (e.g., $p = 0.05$ corresponds to a 5% error). The success rate deteriorates but remains larger than

zero despite large errors (Fig. 5h), showing the robustness of our control framework. However, a 5% error decreases the mean success rate in about 30%, emphasizing the importance of accurately mapping ecological networks for controlling microbial communities.

**Application.** We analyzed the ecological network of the gut microbiota of germ-free mice that were pre-colonized with a mixture of human commensal bacterial type strains and then

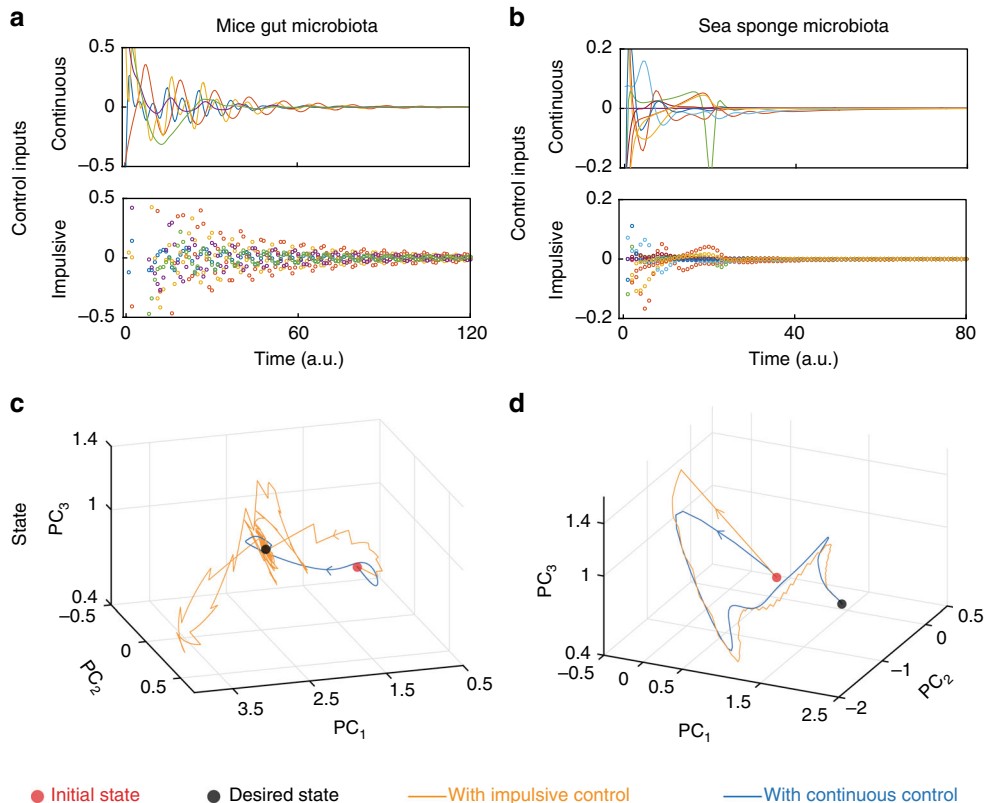

**Fig. 6** Controlling host-associated microbial communities. The controlled population dynamics of both microbial communities were simulated using the controlled GLV equations (see Supplementary Note 7 for details). The intrinsic growth rates were adjusted such that the community has an initial "diseased" equilibrium state $x_0$ in which one species (C. difficile for the mice gut microbiota) is overabundant compared to the rest of species. We chose the desired state $x_d$ as another equilibrium with a more balanced abundance profile. For each microbial community, we used the minimum set of driver species identified in Fig. 3. **a**, **b** Control inputs obtained using the linear MPC for the impulsive and continuous control schemes. **c**, **d** Projection of the high-dimensional abundance profiles (states of the microbial communities) into their first three principal components (PCs). See Supplementary Figure 1 for the temporal response of each species. The calculated control strategies applied to the driver species succeed in driving the community to the desired state, using either continuous or impulsive control

infected with *C. difficile* spores[22]. In Fig. 3a, we identified a minimum set of five driver species in this 14-species community: *Ruminococcus obeum* ($x_1$), *Raphitoma mirabilis* ($x_{12}$), *Bacteroides ovatus* ($x_2$), *Clostridium ramosum* ($x_6$), and *Akkermansia muciniphila* ($x_{10}$). We also used the ecological network underlying the core microbiota of the sea sponge *I. oros*[23], finding ten driver species in this twenty-species community (Fig. 3b).

We studied by simulation the efficacy of the identified driver species and the linear MPC for driving these two microbial communities, assuming that their dynamics are uncertain (see Supplementary Note 7 for details of the simulation). For the mice gut microbiota, our framework succeeds in driving the community from an initial state where *C. difficile* is overabundant towards the desired state with a better balance of species (Fig. 6a, c). Similar results were obtained for controlling the core microbiota of *I. oros* (Fig. 6b, d). These results show again that the linear MPC method is robust enough to drive nonlinear microbial communities.

## Discussion
Our theoretical framework allows systematically and efficiently controlling microbial communities towards desired states by identifying their driver species. Identifying the driver species of a microbial community only requires knowledge of its underlying ecological network. Note that there could be multiple different minimum sets of driver species for the same community. If the cost of choosing any species as a driver species is known, a

combinatorial optimization scheme will allow selecting the best minimum driver species set. We emphasize that the driver species discussed here may not coincide with other notions in ecology such as keystone[47,48] or core[49] species. For example, the selection of driver species do not directly depend on their abundances, while keystone species do[47].

For large uncertain communities, the linear model predictive controller gives a robust and efficient way to calculate the control inputs. The performance of this controller could be further improved by modeling the susceptibility of species to the control actions (e.g., pharmacokinetics). In such case, different control actions could be modeled by different pairs {*f*, *g*}, making the conditions for the absence of autonomous elements different for continuous and impulsive control actions. Control algorithms based on reinforcement learning[50] (RL) could provide even better performance. Our characterization of minimum sets of driver species will help to efficiently apply those control algorithms to microbial communities, as RL algorithms require specifying the "driver variables" they can actuate[51]. Here, controlling small synthetic communities could provide valuable insights for designing such controllers. We also note that altering the ecological network or obtaining a "simplified" network, in the spirit of refs. [52,53], could be complementary control approaches (e.g., for reducing the minimum number of driver species).

It has been suggested that the success of ecosystem management strategies could be predicted using the notion of controllability[54]. However, this notion is somewhat inadequate for

microbial communities and other biological systems. By their nature, biological systems cannot be fully controllable because there are states they cannot reach (e.g., states with negative abundances). Furthermore, since dynamic models for microbial communities are nonlinear and uncertain, it is impossible even to test if those systems are controllable. Structural accessibility overcomes these two limitations, generalizing the notion of accessibility[31] to systems with uncertain dynamics. Counter-intuitively, our mathematical formalism suggests that communities with more complicated population dynamics (i.e., deformations with larger size) require fewer driver species. However, using fewer driver species could complicate the design of control strategies (Remark 9 in Supplementary Note 5). Indeed, by choosing an adequate base model[55] and making mild assumptions on the dynamics (i.e., $f$ and $g$ are meromorphic functions), our framework can identify minimum sets of "driver variables" for general nonlinear systems when their underlying networks are known (see Supplementary Note 9 for an example of a small gene regulatory network).

There are two limitations in our current framework for controlling microbial communities. First, stochastic effects are considered negligible. Incorporating stochastic effects yields stochastic differential equations for which the notion of autonomous elements still needs to be mathematically formulated. We anticipate that this is quite challenging, but definitely merits further studies. Second, our current framework does not explicitly model the dynamics of resources provided to and/or chemicals produced by the microbial species[56–63]. Our characterization of driver species only applies to some instances of resource-based models, e.g., the classical MacArthur's consumer-resource model[64] when the resource dynamics is much faster than the species dynamics[65]. For general resource-based models, identifying their driver species requires analyzing a new kind of "output accessibility" that characterizes the absence of autonomous elements in the species abundances and ignores autonomous elements in the resource abundances. Then, the notion of "structural output accessibility" (i.e., generic output accessibility given an adequate base model) would provide a nonlinear counterpart of linear target controllability[66]. Structural output accessibility could allow us to identify driver species and/or "driver resources" of a community from knowing the bipartite interaction network of species and resources. This is beyond the scope of this work and deserves dedicated efforts.

To fully harvest the benefits of controlling microbial communities, a stronger synergy between microbial ecology and control theory is necessary. We hope that this work will catalyze new interdisciplinary approaches that enhance our ability to control complex microbial communities inside and around us.

## Methods

**Detecting autonomous elements in the continuous control scheme.** For the continuous control systems of Eq. (2), the notion of autonomous elements and the conditions for their absence are well understood, since they define when a system is accessible (see Supplementary Note 2.2 and ref. [31] for details). An autonomous element for Eq. (2) is a non-constant function $\xi(x)$ such that there exists an integer $\nu \geq 0$ and a meromorphic function $F$ such that $F(\xi, \dot{\xi}, \cdots, \xi^{(\nu)}) = 0$. In words, an autonomous element $\xi$ is an "internal variable" of the system that evolves completely unaffected by the control inputs. System (2) is said accessible if it has no autonomous element[31].

The absence of autonomous elements can be characterized by using a mathematical formalism based on differential one-forms[31]. Consider the set of meromorphic functions $\mathcal{K}$ in the variables $\{x, u, \dot{u}, \ddot{u}, \cdots\}$, and the sets of differential symbols $dx = (dx_1, \cdots, dx_N)^\top$ and $du = (du_1, \cdots, du_M)^\top$. Let $\mathcal{X} = \mathrm{span}_\mathcal{K}\{dx\}$ be the vector space spanned over $\mathcal{K}$ by the elements of $dx$, intuitively playing the role of "all functions of state variables". Any $\omega \in \mathcal{X}$ is a "one-form"[31] (see Supplementary Note 2.1 for details). In this setting, the chain rule provides a way to formally operate with one-forms, such as taking time derivatives: if $\omega = \beta^\top dx$ then $\dot{\omega} := \dot{\beta}^\top dx + \beta^\top d\dot{x}$. To identify the presence of autonomous

elements in the dynamics with continuous control of Eq. (2), one calculates the sequence of subspaces $\mathcal{H}_k \subset \mathcal{X}$ defined recursively by

$$\mathcal{H}_k = \{\omega \in \mathcal{H}_k | \dot{\omega} \in \mathcal{H}_k\}, \quad k \geq 1, \tag{7}$$

starting with $\mathcal{H}_1 = \mathcal{X}$. Then, one can prove that Eq. (2) lacks autonomous elements if and only if there exists an integer $k^*$ such that $\mathcal{H}_{k^*} = \{0\}$, see ref. [31] (page 49, Thm.3.17).

**Detecting autonomous elements in the impulsive control scheme.** For the impulsive control systems of Eq. (3) the notions of autonomous elements and accessibility are rather unexplored. Recall that an autonomous element is an internal variable of the system that is completely unaffected by the control actions. To introduce a suitable definition of autonomous element for the impulsive control systems, note that the control inputs cause "jumps" in the actuated variables (i.e., discontinuities). These jumps are propagated to other state variables by the continuous dynamics. Thus, we define an autonomous element of Eq. (3) as a non-constant function $\xi(x)$ such that $\xi(x(t))$, $t \in \mathbb{R}$, is a $\mathcal{C}^\infty$ function (i.e., infinitely differentiable function) under any impulsive input (see Supplementary Note 2.3 for details). By analogy to the case of continuous control, we say that system (3) is accessible if it has no autonomous element according to the above definition.

To characterize the accessibility of impulsive control systems, we built the sequence of subspaces $\mathcal{H}_k$ of all functions of the state variables that can be differentiated at least $(k-1)$ times (see details in Supplementary Note 2.3). The functions belonging to the limit $\mathcal{H}_\infty$ are the autonomous elements of the system, since they are completely unaffected by the control inputs. Consequently, because the limit subspace $\mathcal{H}_\infty$ is also "integrable" (informally, it does not contain "fictitious" autonomous elements), accessibility is equivalent to the condition $\mathcal{H}_\infty = \{0\}$ (see Theorem 2 in Supplementary Note 2). We further prove that the limit $\mathcal{H}_\infty$ is attained in a finite step (i.e., there exists a finite $k^*$ such that $\mathcal{H}_{k^*} = \mathcal{H}_{k^*+1} = \cdots = \mathcal{H}_\infty$).

We illustrate the above formalism using the three-species microbial community of Fig. 1 where $x_3$ is the actuated species (see caption for its population dynamics). To compute the sequence $\mathcal{H}_k$, one starts by definition with $\mathcal{H}_1 = \mathrm{span}_\mathcal{K}\{dx_1, dx_2, dx_3\}$. Next, $\mathcal{H}_2$ are all one-forms in $\mathcal{H}_1$ that can be differentiated once (i.e., they are continuous, so they are not directly affected by $u$). Because $u$ actuates $x_3$, we get $\mathcal{H}_2 = \mathrm{span}_\mathcal{K}\{dx_2, dx_1\}$. Similarly, $\mathcal{H}_3$ are all those one-forms in $\mathcal{H}_2$ that can be differentiated twice (i.e., their first derivative is continuous), yielding $\mathcal{H}_3 = \mathrm{span}_\mathcal{K}\{x_2 dx_1 + x_1 dx_2\}$. Finally, $\mathcal{H}_4 = \{0\}$ (see details in Example 1 in Supplementary Note 2). This implies that the controlled population dynamics is free of autonomous elements and hence it is accessible. See also Example 2 in Supplementary Note 2 for a community with autonomous elements.

**Detecting autonomous elements without knowledge of the population dynamics.** When the controlled population dynamics of the microbial community is unknown, we consider the class $\mathfrak{D}$ of all controlled dynamics that the community may have given we know its controlled ecological network. Identifying the presence of autonomous elements in the full class $\mathfrak{D}$ becomes possible thanks to so-called "generic properties" of meromorphic functions[31]. This is a mathematical property implying that a meromorphic function will satisfy a certain condition in almost all points of its domain—that is, everywhere except for a zero-measure set of "singularities"—provided that such condition holds at a single point. We exploited this property to prove that, generically, increasing the size $C$ of a deformation cannot create new autonomous elements (Proposition 1 in Supplementary Note 3). See Fig. 2c for an illustration. This result allows us to only search for autonomous elements on the subset $\mathfrak{D}_0 \subset \mathfrak{D}$ of all $\{f, g\} \in \mathfrak{D}$ with size $C = 0$, corresponding to all base controlled GLV models of Eq. (4). Finally, we proved that the generic absence of autonomous elements in $\mathfrak{D}_0$ can be determined only from the topology of the controlled ecological network $\mathcal{G}^c$ (Theorem 3 in Supplementary Note 3).

**Identifying a minimum set of driver species.** Let $\tilde{\mathcal{G}}(X)$ be the subgraph obtained by removing all self-loops from the ecological network $\mathcal{G}(X)$ of the (uncontrolled) community. Let $\tilde{\mathcal{B}}(X^- \cup X^+)$ be the bipartite representation of $\tilde{\mathcal{G}}(X)$, built by placing the edge $(x_j^+, x_i^-)$ in $\tilde{\mathcal{B}}$ if the directed edge $(x_j \to x_i)$ is in $\tilde{\mathcal{G}}$. Then, to identify a minimum set of driver species, we applied the notion of a "dedicated input configuration" introduced in ref. [41] (see details in Supplementary Note 4).

A strongly connected component (SCC) is said "non-top linked" if it has no incoming edges from other SCCs. Let $M^*$ be a maximum matching in $\tilde{\mathcal{B}}$. Then, a non-top linked SCC is said to be "top assignable" with respect to $M^*$ if it contains at least one right-unmatched node in $M^*$. Let $Z \subseteq X$ be the set of right-unmatched nodes of some maximum matching of $\tilde{\mathcal{B}}$ with maximum top assignability. Let $W \subseteq X$ be a set consisting of one state node from each non-top linked SCC of $\tilde{\mathcal{G}}$ not already present in $Z$. Then, we prove that $X_D \subseteq X$ is a minimum set of driver species if and only if there exist two disjoint subsets $Z$ and $W$ as defined above, such that $X_D = Z \cup W$ (Proposition 3 in Supplementary Note 4). Using this result, we applied Algorithm 1 of ref. [41] to $\tilde{\mathcal{G}}$ to obtain a minimum set of driver species. This algorithm is implemented in Julia as the

DriverSpecies function in the DriverSpeciesModule package. This algorithm is illustrated for communities of $N = 100$ species in Fig. 5a and Supplementary Fig. 2.

**Choosing the prediction horizon.** To choose the prediction horizon $L$ for the nonlinear MPC we proved there are two possible cases (Theorem 4 in Supplementary Note 5). First, when the community can be driven to $x_d$ using $L < \infty$ impulsive control inputs. Second, when the community can only be asymptotically driven to $x_d$, meaning that $L \gg N$ should be chosen sufficiently large. This second case could be circumvented by increasing the number of actuated species (Remark 8 in Supplementary Note 5).

**Reporting summary.** Further information on experimental design is available in the Nature Research Reporting Summary linked to this article.

**Code availability.** A Julia implementation of the algorithm for identifying a minimum set of driver species, as well as all other functions necessary to reproduce the results of the paper, is provided at the GitHub repository: https://github.com/mtangulo/DriverSpecies.

## Data availability

All the experimental datasets analyzed in this study are publicly available.

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

## Acknowledgements
M.T.A. acknowledges the financial support from CONACyT, Mexico, and L2SN, France. The authors thank Jorge X. Velasco, Jorge Zañudo, Jean-Jacques Slotine, Chuliang Song, and Yandong Xiao for valuable comments and discussions.

## Author contributions
Y.-Y.L. initiated the project. M.T.A. and Y.-Y.L. conceived and designed the project together. M.T.A. and C.H.M. did the theoretical analysis. M.T.A. did the numerical analysis. M.T.A. and Y.-Y.L. wrote the manuscript. C.H.M. edited the manuscript.

## Additional information

**Competing interests:** The authors declare no competing interests.

