## [Peer Review File · Nature Communications]

Reviewers' comments:

Reviewer #1 (Remarks to the Author):

Report on NCOMMS-18-15520

Controlling complex microbial communities: a network-based approach

The paper presents a framework to control the microbial communities using the approach of linear model predictive control (LMPC). The main idea is to identify a minimal set of driver nodes (species) and then to apply a properly chosen, continuous or impulsive control signal to each node in the driver set, so as to make the system evolve from an initial stable steady state to a targeted stable steady state. The way to determine the minimal set of driver species is based completely on network structure: no dynamics are taken into account - the notion of structural accessibility used by the authors. The principle of maximum matching, which was developed for linear networks, was adopted to determine the driver set, and the control signals are obtained from LMPC through solving the optimization problem with a quadratic cost function. For the microbial network, this was done using the linearized version of the generalized Lotka-Volterra (GLV) population dynamics.

The paper is not suitable for publication, because of the following issues.

Major issues

1. **Lack of originality.** The main achievement of the paper was claimed to be the development of a control strategy for microbial communities with nonlinear population dynamics and a large number of species. The authors accomplished this by linearizing the population dynamics and using the linearized system to predict the response of the network to control actions. The idea of controlling a nonlinear network based on linearized dynamics and optimizing some cost function was already published last year by a group from University of New Mexico [Klickstein et al., "Locally optimal control of complex networks," *Phys. Rev. Lett.* **119**, 268301 (2017)]. In fact, this published work was more carefully carried out to address physically significant issues such as the switching of linearized neighborhoods and the associated energy cost, and the method was tested using diverse types of complex networks. Comparing the published PRL work, the present manuscript (see the issues below) appears to be a step back.
2. **Questionable method to determine the driver nodes.** Because of the very special circumstance under which the structural accessibility of a given network is determined, the nodal dynamical processes, e.g., linear, weakly nonlinear or strongly nonlinear, have absolutely no effect on the driver set. That is, one driver set works for all types of dynamics, regardless of whether continuous or impulsive control signals are used. If true, this is at best unreasonable. In fact, this is absurd because the maximum matching method was suitable only for directed, linear networks. It does not work even for linear, undirected networks, let alone nonlinear dynamical networks. Microbial networks are highly nonlinear. There is no valid scientific base to claim that such a nonlinear network can be controlled based on a limited scheme for linear networks.
3. **Non-uniqueness of driver set.** In general, for a given complex network, the linear controllability theory (e.g., based on maximum matching) would give a unique number of driver

nodes. However, the set of driver nodes can be far from being unique. As the size of the network is increased, the number of possibilities for the driver set grows exponentially. One can even calculate the probability for each node to be a driver. The authors used the maximum matching theory to determine a single set of driver nodes and claimed it to be optimal for highly nonlinear networks, which seems quite unreasonable.

Technical issues

- (a) The goal of this paper is to control the complex microbial community systems that live in a noisy environment. The issue of the effects of noise on control is important, but it was completely ignored in the present manuscript.
- (b) On pages 16-17 in the main text, the authors stated: “Next, by by rewriting the controlled population dynamics of the community as $\{f(x), g(x)\} = \{\hat{A}x + \omega_x, \hat{B} + \omega_\mu\}$, we use the pair $\{\hat{A}x, \hat{B}\}$ to provide a linear prediction for the response of the community to control inputs. Here, the linear functions $(\omega_x, \omega_\mu) = (f - \hat{A}x, g - \hat{B})$ represent perturbations whose magnitude depend on how well the linear pair $\{\hat{A}x, \hat{B}\}$ approximates the true dynamics $\{f(x), g(x)\}$ of the community. . . . The above linear MPC has several other advantages: it requires minimal knowledge of the controlled population dynamics of the community; it is robust to the perturbations (ω_x, ω_μ) ; and it is allows calculating the control signals for the continuous control scheme.”

There is no evidence that the linear MPC is robust to the perturbations in (ω_x, ω_μ) . For the variational model on page 10 in the main text, to what extent are the predictions from the linearized systems applicable to the actual, highly nonlinear dynamical system? How do the perturbations affect the result in Fig. 4 in the main text?

- (c) In supplementary Note 6, Remark 9, the authors stated “In general, the linear MPC is guaranteed to succeed only if the desired state is ‘close enough’ to the initial state. . . . how ‘close’ or ‘far’ is a desired state depends on how well the linear dynamics approximates the true population dynamics of the community.”

The authors claimed in the main text that the control strategies are applicable to microbial communities with uncertain population dynamics and a large number of species. A critical technical issue that was not addressed is actually how “close” it is necessary to drive the system to some linearized neighborhood for the control to be effective, especially when there are uncertainties in the dynamical equations and the system is high dimensional (due to the large number of species).

- (d) On page 11 in the main text, the authors stated “*i*-th species is actuated by the α -th control input.” It appears that the so-stated α -th control input was not specified.
- (e) The authors did not provide a description of the gray dots in panels (b) and (d) in Fig. 3. Are these stable steady states or something else?

- (f) On page 15 in the Supplemental Materials, there is an error in the equation $\int_{x_{20}}^{x_2^{(t_1)}} \frac{dx_2}{x_2} = \int_0^{t_1} [1 - x_2(t)] dt$. The correct one should be $\int_{x_{20}}^{x_2^{(t_1)}} \frac{dx_2}{x_2} = \int_0^{t_1} [1 - x_3(t)] dt$.

Reviewer #2 (Remarks to the Author):

This paper presents a very promising approach to the problem of how to understand, predict and control the behaviour of complex ecological networks. It presents an approach and some case studies for testing it.

I see very promising results giving high hopes for ecological applications - but I also believe that a mass amount of further tests are still needed (in silico and in vitro) in order to make this really convincing.

Comments:

The first sentence of the abstract is a vague overstatement, while the last one is quite speculative.

Key microorganisms can be identified based on network topology in a structural sense (see <https://www.frontiersin.org/articles/10.3389/fmicb.2014.00219/full>, <https://www.nature.com/articles/srep15920>). To infer dynamical behaviour (not necessarily controlling, even predicting or only understanding) is a massive challenge. I missed an overview (kind of metaanalysis) of earlier papers and the comparison of earlier results to the ones of this approach.

Finding "core" sets of organisms is a very nice idea and very promising (see <https://journals.aps.org/pre/abstract/10.1103/PhysRevE.65.026103>, <https://www.sciencedirect.com/science/article/pii/S1470160X1730359X>). This multi-species view may be based only on network structure or dynamics. To what extent the results can be generalizable here?

In an ecological (but also biological) context, the word "control" sounds very positivist. It is used for "biological control" (see https://link.springer.com/chapter/10.1007%2F1-4020-4767-3_7) but also these simplistic approaches bring lots of surprises. In multi-species, multi-parameter, highly non-linear ecodynamics, this seems to be not easily predictable by network structure. So, I disagree with the statement "The difficulty in this second challenge originates in our insufficient knowledge of the microbial dynamics and their interactions", I think this is inherent, not knowledge-limited.

"disruption to the gut microbiota" and "driving ... back to their healthy stage" are to be discussed more carefully, given the extreme spatial, temporal and individual-level variability of these microbes and ALSO their systems. They are in continuous change and any "disruption" is much more like a change of some trajectory than "loosing a part" or something like this. This might be better discussed and presented.

"Note that this ecological network is fundamentally different from correlation or co-occurrence networks, because those networks are undirected." - there are differences from the viewpoint of symmetry and direction, this is clear, but I would not say "fundamentally". The association-based, statistical interaction networks are quite good proxies for "real" interaction networks. Most interactions are bidirectional (even if not symmetrical), so the effects can be considered undirected (if we do not consider the signs). The problem is that one cannot construct complex microbial networks without these proxies, see the methods mentioned just in the followings.

"But that would be overkill" is unclear to me

Figure 2b: the second control input is on x_1 , not on x_2

Selecting driver species for control can be problematic since there is an interplay between network

topology and actual abundance values. Highly abundant organisms can be popular partners (prey or other). This adds another level of complexity, I think. Can this be treated simply and efficiently?

Figure 4: we see 5 drivers for 9 non-drivers (mice) and 10 drivers for 10 non-drivers (spoge), I hope I counted well, and the non-drivers are typically perpheric, poorly connected nodes. So, almost all well-connected node needs to be directly controlled in order to indirectly control roughly the same number of poorly connected nodes - I am not sure this is technically feasible or plausible, given the hundreds of microbial species in realistic (really complex) communities.

"we found that the conditions for the absence of autonomous elements for the continuous and the impulsive control schemes are identical" - differences probably emerge if more complicated functional response kinetics are considered with saturation terms and spatial constraints.

"Note that once G_c is structurally accessible, this condition will not be violated if new edges are added to the network" - this is an important finding, but how sensitive is the identification of drivers to losing edges?

I think it is a major question how can controllability depends on the two states (initial and desired). If the initial state would naturally evolve towards the desired one, or just in the contrary direction, might be crucial here. So, instead of looking at trajectories between two points, it would be nice to better discuss the positions of these two points in the landscape.

Reviewer #3 (Remarks to the Author):

Dear Editor,

In ms NCOMMS-18-15520, the authors present a powerful method for controlling the dynamics of microbial communities, given relatively little information about the nature of their interactions. Importantly, the methodology is applicable to cases where the dynamics are nonlinear, in contrast the "structural controllability" which is currently a very hot topic in network science.

The work appears to be without technical error and the manuscript is mostly clear, though the work is technically dense and will likely be challenging for some readers who would otherwise be interested in the work. My specific comments, below, are mostly aimed toward increasing the readability of the work (see especially comment 2, where I encourage the inclusion of more detail in the main text).

I hope these comments are useful to the authors as they revise their ms.

Comment 1

It seems to me that the two problems mentioned at the top of page 3 (knowing the set of minimal species necessary for control and then knowing what that control is) are co-implicated to some degree. It may be prudent for the authors to make this point clear.

Comment 2

Near the bottom of page 7, the authors define the term "autonomous element" and provide an example (Figure 2) explaining the term. However, the authors do not provide an explanation in terms of the topology of the network for why it is that node x3 cannot (by itself) be a driver species, and instead refer the reader to a number of supplementary notes.

It would be very useful if the authors could provide an intuitive explanation for why is it that controlling node x3 confines the state of the network to a low-dimension manifold, then refer the reader to the supplementary notes for additional details.

Comment 3

Typo, line 11 of page 11 ('\alpha' -> j, if I read the authors correctly)

Comment 4

More justification/explanation should be given for the choice of A^j given on line 14 of page 17.

Comment 5

The authors may be interested in some broadly related work that also examines the topology of interactions networks to characterize and/or control network dynamics. I stress the authors needn't feel compelled to cite either of these works:

DOI: 10.1063/1.4809777

DOI: 10.1186/1752-0509-8-53

Reviewer #4 (Remarks to the Author):

Marco Tulio Angulo and colleagues describe in their manuscript 'Controlling complex microbial communities: a network based approach' their strategy on how to access microbial communities through precise manipulation. To do so the authors describe their approach of a control framework based on the notion of structural accessibility. Through this approach they claim their ability to identify 'driver species' through which manipulation becomes feasible.

General comments:

I very much like the authors concept since their method is practical, straight forward and a tangible application of network analysis which has remained too descriptive in many previous studies. Despite the great potential of the described method, experiments for validation provided by the authors are only run in simulations and only for a limited number of nodes. Of course it would have been amazing if the authors could have proven their hypothesis through real control experiments: for example by manually increasing / decreasing *Clostridium difficile* abundance in a lab experiment. Alternatively there are data-sets of community disturbance available in the literature. It would have been great to see if the MPC model correctly predicts the network rewiring.

As mentioned above, I find the paper sound and novel. The authors, however, are sparse on highlighting the novelty of their approach in the paper. It would be good to highlight in more detail the novelty compared to available approaches. I see the novelty in the accessibility and controllability of the approach.

The main claim of the paper that is to develop a reliable method to identify driver nodes for manipulation of large networks has been confirmed and proven in theory very elegantly, the paper, however, lacks experimental confirmation which could be by using available datasets. Unfortunately simulations do not take into consideration large networks although the authors write about large networks. Networks from natural samples (or ecological networks) contain rarely below 100 nodes and therefore I would not consider a network with < 100 nodes a large network.

Concrete suggestions:

- 1) The authors should certainly re-think the term and usage of 'ecological networks'. They could just tone down the text and re-write those parts.
- 2) The authors should certainly check available data-sets and use 'real' data examples to show that the model is trained and deployed correctly.

Some more detailed comments:

The main criticism is that the authors take a number of assumptions into account which can be hardly justified:

A) The Assumption that the starting network is directed: in very few cases in ecology or in microbiology there is enough data available to infer a directed network. With metagenomics data even more. I therefore challenge this assumption.

B) Driver species are definitely the way to go to gather more insights in network analysis. However, for intrinsic properties of networks, driver nodes correlate positively with the heterogeneity and density of the network. This means that identifying the driver nodes is relevant only if the network is dense enough. The authors do not make any assumption on network homogeneity and density. On the contrary at pg 19 line 4 they state that they found 10 driver nodes out of 20. This sounds like not a really convenient strategy in this case.

Response to Reviewer #1

The paper presents a framework to control the microbial communities using the approach of linear model predictive control (LMPC). The main idea is to identify a minimal set of driver nodes (species) and then to apply a properly chosen, continuous or impulsive control signal to each node in the driver set, so as to make the system evolve from an initial stable steady state to a targeted stable steady state. The way to determine the minimal set of driver species is based completely on network structure: no dynamics are taken into account - the notion of structural accessibility used by the authors. The principle of maximum matching, which was developed for linear networks, was adopted to determine the driver set, and the control signals are obtained from LMPC through solving the optimization problem with a quadratic cost function. For the microbial network, this was done using the linearized version of the generalized Lotka-Volterra (GLV) population dynamics.

We thank the Reviewer #1 for reviewing our paper. Below we provide a point-by-point response to her/his comments.

The paper is not suitable for publication, because of the following issues.

Major issues

1. Lack of originality. The main achievement of the paper was claimed to be the development of a control strategy for microbial communities with nonlinear population dynamics and a large number of species. The authors accomplished this by linearizing the population dynamics and using the linearized system to predict the response of the network to control actions. The idea of controlling a nonlinear network based on linearized dynamics and optimizing some cost function was already published last year by a group from University of New Mexico [Klickstein et al., “Locally optimal control of complex networks,” *Phys. Rev. Lett.* 119, 268301 (2017)]. In fact, this published work was more carefully carried out to address physically significant issues such as the switching of linearized neighborhoods and the associated energy cost, and the method was tested using diverse types of complex networks. Comparing the published PRL work, the present manuscript (see the issues below) appears to be a step back.

We appreciate the opportunity to clarify the originality of our work. We think Reviewer #1 might have completely misunderstood our paper. This gives us the opportunity to stress some of the main messages.

First of all, to identify the driver species of a microbial community, our framework **does not rely on the linearization of its population dynamics at all**. Rather, our framework is based on analyzing the accessibility of the population dynamics —namely, the absence of autonomous elements (see Fig. 2 and the definition in Page 8 of the main text). We emphasize that the **accessibility of a nonlinear controlled system is necessary but not sufficient for the controllability of its linearization**. To prove this point, let’s just consider the following simple example that is accessible and (nonlinearly) controllable, but its linearization at the origin is not controllable:

$$\dot{x}_1 = x_2^3, \quad \dot{x}_2 = u.$$

As described in SI Sec 2.3, calculating the sequence of subspaces $\mathcal{H}_2 = \text{span}_{\mathcal{X}}\{dx_1\}$, and $\mathcal{H}_3 = 0$ proves the above system is accessible. This implies that its state is not constrained to any low dimensional manifold. Indeed, it is not hard to see that the system can be steered to any point of \mathbb{R}^2 .

Second, to calculate a control signal that drives the microbial community to the desired state, we first consider a **nonlinear** impulsive Model Predictive Control (MPC), which takes full advantage of the precise knowledge of the nonlinear population dynamics and **does not rely on its linearization** (see Eq. (6) of the main text and Fig.3a,b). When the precise knowledge of the nonlinear population dynamics is not available (but the structure of the ecological network is known), we consider a linear impulsive MPC method based on a proxy (\hat{A}, \hat{B}) of the interaction and susceptibility matrices of the microbial community. **We emphasize that in our method the pair (\hat{A}, \hat{B}) does not need to be any linearization of the population dynamics.** We illustrate this point in Fig.3c,d of the main text, using

$$\hat{A} = \begin{bmatrix} -0.5 & 0 & -0.1 \\ 0 & -5 & 1 \\ 0 & 0 & -1 \end{bmatrix}$$

as a proxy for the interaction matrix of the microbial community, which is completely different from the linearization of the population dynamics at the desired state:

$$\begin{bmatrix} -0.37 & 0 & -0.05 \\ 0 & -5.31 & 0.52 \\ 0 & 0 & -1 \end{bmatrix}.$$

Third, when the precise knowledge of the nonlinear population dynamics is not available (which is often the case for complex microbial communities), it is not possible to calculate its linearization either. Consequently, it is impossible to directly apply the method of Klickstein *et al.* suggested by Reviewer #1. By contrast, as we mention above and illustrated in Fig. 3c,d of our paper, our method can still be applied in this case, as long as the structure of the ecological network of the microbial community is known. **In this sense, we consider our work is a big advance over the method of Klickstein *et al.*, rather than a step back.**

Finally, to help Reviewer #1 better understand the originality of our work, we list our contributions as follows:

- 1) We introduce a new definition of *autonomous element* for impulsive control systems (Definition 3 in Supplementary Note 2), which has never been defined in the literature.
- 2) We characterize both necessary and sufficient conditions for the absence of autonomous elements in arbitrary nonlinear impulsive control systems (Theorem 2 in Supplementary Note 2). This can be used to better model certain control actions applied to microbial communities.
- 3) We introduce the notion of *structural accessibility* as: (i) a generalization of the notion of accessibility to uncertain nonlinear systems; and (ii) as a generalization of the notion of (linear) structural controllability to nonlinear systems. To define this notion, we also introduce the notions of a *base model* (the “simplest” dynamics that the microbial community may have, chosen as the Generalized Lotka-Volterra model) and their deformations (i.e., the more “complex” dynamics that the community may have).

- 4) We provide a complete *graph theoretical* or *network characterization* of structural accessibility (Theorem 3 in Supplementary Note 3). This characterization relies on proving that increasing the complexity of a deformation cannot (generically) invalidate a set of driver species (Proposition 1 in Supplementary Note 3).
- 5) Based on the above characterization of structural accessibility, we introduce a method to identify a minimal set of driver species of a microbial community solely from the topology of its ecological network (Supplementary Note 4).
- 6) When the population dynamics of the community is known, we construct an impulsive nonlinear Model Predictive Controller (MPC) to generate the control signal that needs to be applied to the driver species to steer the community towards a desired state (see Eq. (6) and Fig. 3a,b of the main text).
- 7) To actually implement the above impulsive nonlinear MPC controller, we built a mathematically rigorous method to calculate a sufficient number of impulses needed to drive the community towards the desired state (Theorem 4 in Supplementary Note 4).
- 8) When no detailed knowledge of the population dynamics of the community is available, we proposed a linear MPC based on a proxy of the interaction matrix and susceptibility matrix of the community. We demonstrate that this approach can succeed in steering microbial communities towards desired states, despite of the uncertain nonlinear population dynamics (Fig. 3c,d). In the revised main text, we include a detailed validation of the linear MPC applied to the driver species for controlling large microbial communities ($N = 100$ species) with nonlinear population dynamics and diverse ecological network parameters (Fig. 4).
- 9) We illustrate by simulation the application of our framework in controlling the gut microbiota of mice infected with *C. difficile*, and the core microbiota of the sea sponge *Ircinia oros*, without using detailed knowledge of their population dynamics.
- 10) We illustrate the potential application of our framework to control other biological systems, e.g., the Repressilator (a synthetic genetic regulatory network, see Supplementary Figure 1).

Taken together, we believe the above contributions represent significant advances in controlling complex networked systems by introducing and characterizing the notion of structural accessibility for nonlinear systems with impulsive or continuous control. In particular, we demonstrated the application of our control theoretical framework in the field of controlling microbial communities, by providing a systematic pipeline to efficiently steer large microbial communities towards desired states.

To address Reviewer #1's concern, in the revised manuscript, we have added the following remark to Supplementary Note 6:

Remark 13. In Ref. [35], the authors proposed a strategy to optimally control complex networks based on the linearization of their nonlinear dynamics. To apply this approach for controlling microbial communities, it would be necessary to have exact knowledge of their population dynamics in order to calculate the linearization. Our approach circumvents this limitation, requiring only to know a proxy (\hat{A}, \hat{B}) of the interaction matrix and susceptibility matrix of the community. We emphasize that (\hat{A}, \hat{B}) does not need to coincide with the linearization of the dynamics (see, e.g., Fig. 3c,d of the main text).

The above paragraph explicitly clarifies the difference between our framework and the method of Klickstein *et al.* (ref. [35] of the Supplementary Notes).

2. Questionable method to determine the driver nodes. Because of the very special circumstance under which the structural accessibility of a given network is determined, the nodal dynamical processes, e.g., linear, weakly nonlinear or strongly nonlinear, have absolutely no effect on the driver set. That is, one driver set works for all types of dynamics, regardless of whether continuous or impulsive control signals are used. If true, this is at best unreasonable. In fact, this is absurd because the maximum matching method was suitable only for directed, linear networks. It does not work even for linear, undirected networks, let alone nonlinear dynamical networks. Microbial networks are highly nonlinear. There is no valid scientific base to claim that such a nonlinear network can be controlled based on a limited scheme for linear networks.

We appreciate the opportunity to clarify this point.

First of all, we fully agree with Reviewer #1 that the set of driver species of a microbial community depends on its specific population dynamics. Indeed, to illustrate this point, we prepared Fig. 2a,d in the main text to show two communities with different population dynamics but identical ecological networks. Using our mathematical formalism, we show species x_3 is a driver species for the first community but **not** for the second community. In general, it is not obvious if it is even possible to determine a set of driver species that “works for all types of dynamics”, especially when those dynamics can be nonlinear. As an original contribution, our work presents a mathematically rigorous way to circumvent such a seemingly unavoidable limitation. Specifically, our solution is based on introducing the notions of “base model” (here chosen as the Generalized Lotka-Volterra model) and their “deformations” (Definition 5 of Supplementary Note 3). The base model captures the “simplest” population dynamics that the community can have. The deformations of the base model capture the other “more complex” population dynamics that the community may exhibit (see Eq.(5) of the main text). Within this framework, we can prove that increasing the complexity of a deformation cannot (generically) invalidate a given set of driver species (Proposition 1 in Supplementary Note 3). **In other words, if a set of driver species is identified using the base model, then the same set of driver species will “work” for almost all dynamics with higher complexity (i.e., for almost all deformations).** We illustrate this result in Fig. 2d of the main text, where increasing the complexity of the population dynamics of the community makes x_3 a solo driver species. This key result enabled us to find the driver species of a community from its ecological network under the assumption that the “simplest” dynamics it can take is given by the Generalized Lotka-Volterra model.

Second, we agree with Reviewer #1 that, *a-priori*, there is no reason to even expect that the notion of structural controllability —developed by C.-T. Lin *et al.* **for linear systems** and later applied to complex networks by Y.-Y. Liu and others— can work for **nonlinear systems**. This is actually a key motivation of our work. Indeed, for a very long time, the control theory community has appreciated that different tools are needed to control and analyze nonlinear systems because their behavior is naturally broader, and linear systems are just a very singular subclass in the class of nonlinear systems. This prompted us to develop the new notion of **structural accessibility** as a nonlinear generalization of (linear) **structural controllability**. This is a non-obvious achievement as several preconceived ideas are contradicted with each other. We emphasize that accessibility is

a keystone concept in nonlinear control theory, playing the role of a nonlinear analogue of controllability that is very amenable for analysis (see, e.g., Ref. [28] of the main text). As an original contribution, our paper presents a graph theoretical characterization of necessary and sufficient conditions that render a nonlinear system structurally accessible. In fact, we provided such conditions of structural accessibility in the form of a Theorem with a rigorous mathematical proof (see Theorem 3 of Supplementary Note 3), forming the scientific basis of our control framework for microbial communities. This should have already addressed Reviewer #1's concern that "no valid scientific base" sustains our framework.

3. Non-uniqueness of driver set. In general, for a given complex network, the linear controllability theory (e.g., based on maximum matching) would give a unique number of driver nodes. However, the set of driver nodes can be far from being unique. As the size of the network is increased, the number of possibilities for the driver set grows exponentially. One can even calculate the probability for each node to be a driver. The authors used the maximum matching theory to determine a single set of driver nodes and claimed it to be optimal for highly nonlinear networks, which seems quite unreasonable.

We appreciate the opportunity to clarify this point. We totally agree with Reviewer #1 that, in general, a minimal set of driver species is not unique. Yet, we don't consider this as an "issue" or disadvantage for our framework for controlling microbial communities. Instead, we consider this as a big advantage, because different sets of driver species provide us more options to choose the control actions that should be applied to drive the microbial community (i.e., which specific prebiotics, probiotics, bacteriostatic agents, or transplantations should be applied).

To address Reviewer #1's concern, in the revised manuscript we have added the following sentence to the Discussion section.

In this paper, we used a maximum matching based algorithm to identify a minimum set of driver species from the ecological network of a given microbial community. In principle, there could be multiple maximum matchings associated with the same network, rendering potentially different minimum sets of driver species. Note that those minimum driver species sets share the same cardinality. We claim that a minimum set of driver species is optimal only in the sense that its cardinality is minimal. If the cost of choosing any species as a driver species is known, one can develop a combinatorial optimization scheme to further pick up the best driver species set. But we feel this is beyond the scope of the current work and hence leave it for future work.

Technical issues

- a) The goal of this paper is to control the complex microbial community systems that live in a noisy environment. The issue of the effects of noise on control is important, but it was completely ignored in the present manuscript.

We thank Reviewer #1 for this very insightful comment. We acknowledge that the applicability of our control framework is limited to those microbial communities where stochastic effects can be neglected. Indeed, we made this assumption explicit as Assumption 1d in Supplementary Note 1. Yet, microbial communities can respond to alterations similar to the control actions we consider

in a strongly deterministic manner (e.g., changes in the concentration of antibiotics). Papers reporting such deterministic behavior include:

- Zhang, Qiucen, Guillaume Lambert, David Liao, Hyunsung Kim, Kristelle Robin, Chih-kuan Tung, Nader Pourmand, and Robert H. Austin. "Acceleration of emergence of bacterial antibiotic resistance in connected microenvironments." *Science* 333, no. 6050 (2011): 1764-1767.
- Vanwonterghem, Inka, Paul D. Jensen, Paul G. Dennis, Philip Hugenholtz, Korneel Rabaey, and Gene W. Tyson. "Deterministic processes guide long-term synchronised population dynamics in replicate anaerobic digesters." *The ISME journal* 8, no. 10 (2014): 2015.
- Frenzt, Zak, Seppe Kuehn, and Stanislas Leibler. "Strongly deterministic population dynamics in closed microbial communities." *Physical Review X* 5, no. 4 (2015): 041014.
- Kent, Angela D., Anthony C. Yannarell, James A. Rusak, Eric W. Triplett, and Katherine D. McMahon. "Synchrony in aquatic microbial community dynamics." *The ISME journal* 1, no. 1 (2007): 38.
- Gobet, Angélique, Simone I. Böer, Susan M. Huse, Justus EE Van Beusekom, Christopher Quince, Mitchell L. Sogin, Antje Boetius, and Alban Ramette. "Diversity and dynamics of rare and of resident bacterial populations in coastal sands." *The ISME journal* 6, no. 3 (2012): 542.

We also emphasize that deterministic mathematical models are frequently used to analyze complex microbial communities, as illustrated in the following influential references:

- Katharine Z Coyte, Jonas Schluter, and Kevin R Foster. The ecology of the microbiome: Networks, competition, and stability. *Science*, 350(6261):663–666, 2015.
- Friedman, Jonathan, Logan M. Higgins, and Jeff Gore. "Community structure follows simple assembly rules in microbial microcosms." *Nature ecology & evolution* 1, no. 5 (2017): 0109.
- Faust, Karoline, and Jeroen Raes. "Microbial interactions: from networks to models." *Nature Reviews Microbiology* 10, no. 8 (2012): 538.
- Buffie, Charlie G., Vanni Bucci, Richard R. Stein, Peter T. McKenney, Lilan Ling, Asia Gobourne, Daniel No et al. "Precision microbiome reconstitution restores bile acid mediated resistance to *Clostridium difficile*." *Nature* 517, no. 7533 (2015): 205.
- Bucci, Vanni, Belinda Tzen, Ning Li, Matt Simmons, Takeshi Tanoue, Elijah Bogart, Luxue Deng et al. "MDSINE: Microbial Dynamical Systems INference Engine for microbiome time-series analyses." *Genome biology* 17, no. 1 (2016): 121.
- Panikov, N. S. "Understanding and prediction of soil microbial community dynamics under global change." *Applied Soil Ecology* 11, no. 2-3 (1999): 161-176.

In summary, our assumption of negligible stochastic effects is both feasible in some experimental situations and frequently adopted in many influential references. Our work provides the ground to develop a mathematically more comprehensive solution to the problem of controlling microbial communities. This will trigger more research activities in this exciting new area.

To address Reviewer #1's concern and clarify the contribution and limitations of our framework, we have added the following sentence to the Discussion section of the revised manuscript:

Note also that in our deterministic framework we don't consider the effects of stochasticity due to, e.g., immigration in microbial communities. From a theoretical viewpoint, incorporating stochastic effects into the model will turn Eqs. (2) and (3) into controlled stochastic differential equations, which are the material of a different scientific area. To the best of our knowledge, the characterization of the accessibility properties of those class of equations remains an open problem and their analysis become intractable in practice. Indeed, the very notion of an autonomous element—the basis for the concept of accessibility— would need to be reformulated. We consider this is beyond the scope of the current work and call for research activities of the control theory community in this area.

- b) On pages 16-17 in the main text, the authors stated: *“Next, by rewriting the controlled population dynamics of the community as $\{f(x), g(x)\} = \{\hat{A}x + w_x, \hat{B} + w_u\}$, we use the pair $\{\hat{A}x, \hat{B}\}$ to provide a linear prediction for the response of the community to control inputs. Here, the linear functions $(w_x, w_u) = (f - \hat{A}x, g - \hat{B})$ represent perturbations whose magnitude depend on how well the linear pair $\{\hat{A}x, \hat{B}\}$ approximates the true dynamics $\{f(x), g(x)\}$ of the community. . . . The above linear MPC has several other advantages: it requires minimal knowledge of the controlled population dynamics of the community; it is robust to the perturbations (w_x, w_u) ; and it allows calculating the control signals for the continuous control scheme.”*

There is no evidence that the linear MPC is robust to the perturbations in (w_x, w_u) ...

We thank Reviewer #1 for this comment, and we appreciate the opportunity to clarify this point. A direct and immediate evidence of the robustness of the proposed linear MPC was provided in Fig. 3d of our main text, where we show that the proposed linear MPC successfully drives a community to the desired state despite having nonlinear population dynamics (see caption of Fig. 1b). Note that for this example the perturbations (w_x, w_u) are not identically zero because the population dynamics of the community is not exactly given by the pair $\{\hat{A}x, \hat{B}\}$.

More generally, as described in page 18 of the main text and Supplementary Note 6, we emphasize that the solution to our linear MPC coincides with the solution to a special Linear Quadratic Regulator (LQR) problem (see Eq. (S19) of Supplementary Note 6). This implies that the proposed linear MPC inherits the robustness properties of a LQR controller. The robustness properties of the continuous time LQR controller (such as 50% gain reduction tolerance, $+\infty$ gain margin, and phase margin of 60°) were established in the 70's in a series of seminal papers by Safonov, Athans, Rosenbrock and others, summarized in books such as: Anderson, Brian DO, and John B. Moore. *Optimal control: linear quadratic methods*. Courier Corporation, 2007. The analysis of the robustness of discrete time LQR controllers—which is obtained in the case of using the proposed impulsive linear MPC— can be found in similar seminal contributions from the 80's such as: Shaked, U. "Guaranteed stability margins for the discrete-time linear quadratic optimal regulator." *IEEE Transactions on Automatic Control* 31, no. 2 (1986): 162-165.

Considering Reviewer #1's comment, we have included the following sentence in page 18 of the revised main text:

[The linear MPC] is robust to the perturbations (w_x, w_u) and other uncertainties (Remark 12 in Supplementary Note 6).

We also included the following remark in the Supplementary Note 6:

Remark 12. Since the linear impulsive and continuous MPCs we proposed here coincide with the solution to discrete and continuous LQR problems, respectively, our proposed controllers will naturally inherit the robustness properties of LQRs. The excellent robustness of continuous time LQRs (such as 50% gain reduction tolerance, $+\infty$ gain margin, and 60 degrees of phase margin) were derived in the 70's in a series of now classical papers by Safonov, Athans, Rosenbrock and others, summarized in books such as [33]. Similarly, the robustness properties of discrete time LQRs were derived in seminal contributions from the 80's such as [34].

Together, we have justified our claim that the proposed linear MPC is robust by describing its connection to the well-known robustness properties of LQR controllers.

Additionally, regarding the illustration of the robustness of the linear MPC in Fig. 3d, we have also included the following sentence in the revised main text (page 19):

The success of the linear MPC in driving a community with nonlinear population dynamics illustrates the robustness of the MPC strategy, since the controller succeeds despite having non-zero perturbations (w_x, w_u) .

Finally, in the revised main text we also present a systematic numerical validation of the linear MPC for controlling large microbial communities with nonlinear population dynamics (specifically, with population dynamics with Holling Type II functional response). As shown in Fig. 4 of the main text, these results corroborate the robustness of the proposed linear MPC, in the sense that the linear MPC can succeed in driving large microbial communities despite having nonlinear population dynamics.

...For the variational model on page 10 in the main text, to what extent are the predictions from the linearized systems applicable to the actual, highly nonlinear dynamical system?...

We thank Reviewer #1 for this insightful comment. We recall that an important property of the solution to LQR problems with infinite horizons is that in such situation the optimal control strategy only depends on the current state (i.e., they are static feedback controllers). This property implies that our linear MPC control with infinite prediction horizon (i.e., with $L \rightarrow \infty$) actually requires **no** explicit prediction of future states of the community, because it only depends on the current state of the community $x(t_k)$ (see Eq. (S20) of Supplementary Note 6).

To address Reviewer #1's comment, we have revised Supplementary Note 6 as follows:

...In the particular case of an infinite prediction horizon $L \rightarrow \infty$, this implies that the solution to the optimization problem of Eq. (S17) takes the form of the linear feedback controller $u(t_k) = K(\hat{x}(t_k) - x_d)$ where $K \in \mathbb{R}^{M \times N}$ is the gain matrix. Since we can

measure the current state of the community, the predicted value of the current state $\hat{x}(t_k)$ simply coincides with the current state $x(t_k)$, giving the final form for the controller:

$$u(t_k) = K(x(t_k) - x_d).$$

where we clarify that in the case of an infinite prediction horizon the linear MPC becomes a static feedback controller that only depends on the current state of the community.

...How do the perturbations affect the result in Fig. 4 in the main text?

The perturbations (w_x, w_u) in Fig. 4 of the original main text make that the actual system trajectory does not necessarily follow the predicted trajectory, but a different one determined by the nonlinear population dynamics (given by the GLV equations described in Supplementary Note 7).

- c) In supplementary Note 6, Remark 9, the authors stated “*In general, the linear MPC is guaranteed to succeed only if the desired state is ‘close enough’ to the initial state. ... how ‘close’ or ‘far’ is a desired state depends on how well the linear dynamics approximates the true population dynamics of the community.*” The authors claimed in the main text that the control strategies are applicable to microbial communities with uncertain population dynamics and a large number of species. A critical technical issue that was not addressed is actually how “close” it is necessary to drive the system to some linearized neighborhood for the control to be effective, especially when there are uncertainties in the dynamical equations and the system is high dimensional (due to the large number of species).

We thank Reviewer #1 for this very legitimate concern. To address this point, in the revised main text we present a detailed in-silico analysis of the success rate of the linear MPC method as a function of the distance of the initial state from the desired state (see Fig. 4 in the main text). Specifically, we considered large random microbial communities of $N = 100$ species with nonlinear population dynamics with Holling Type II functional response. Then, the success rate of the linear MPC was analyzed as a function of the distance to the desired state in three different scenarios:

1. Changes in the parameters of the underlying ecological network of the community (i.e., changes in its connectivity and strength of the inter-species interactions).
2. Changes in the proportion of species that are chosen as driver species.
3. Errors in the ecological network used to identify the driver species and build the linear MPC controller, quantified by a “rewiring probability”.

Our numerical analysis shows that, for distances to desired state ≤ 5 (measured in $\|\cdot\|_2$ -norm), the linear MPC has a high success rate (>0.8) if the connectivity of the ecological network is low (≤ 0.029), or the inter-species interaction strength is small (≤ 1). This new result provides a quantification of “how close” needs to be the desired state from the initial state. The main text has been revised accordingly to incorporate this new result.

- d) On page 11 in the main text, the authors stated “*i-th species is actuated by the α -th control input.*” It appears that the so-stated α -th control input was not specified.

We thank Reviewer #1 for pointing out this typo. It has been corrected in the revised manuscript.

- e) The authors did not provide a description of the gray dots in panels (b) and (d) in Fig. 3. Are these stable steady states or something else?

We thank Reviewer #1 for this comment. Those gray dots are the equilibria of the population dynamics. We have revised the main text to clarify this point.

- f) On page 15 in the Supplemental Materials, there is an error in the equation $\int_{x_{20}}^{x_2(t_1)} \frac{dx_2}{x_2} = \int_0^{t_1} [1 - x_2(t)] dt$. The correct one should be $\int_{x_{20}}^{x_2(t_1)} \frac{dx_2}{x_2} = \int_0^{t_1} [1 - x_3(t)] dt$.

We thank Reviewer #1 for pointing out this typo. It has been corrected in the revised manuscript.

Finally, we thank again Reviewer #1 for her/his very insightful and constructive comments. We hope our responses and revisions have clarified the originality of our contributions in a satisfactory manner.

Response to Reviewer #2

This paper presents a very [p]romising approach to the problem of how to understand, predict and control the behaviour of complex ecological networks. It presents an approach and some case studies for testing it.

I see very promising results giving high hopes for ecological applications - but I also believe that a mass amount of further tests are still needed (in silico and in vitro) in order to make this really convincing.

We thank Reviewer #2 for her/his overall positive assessment of the potential applications of our work. Considering Reviewer #2's concern that "further tests are still needed", in the revised main text we provide a detailed in-silico validation of our control framework for large microbial communities of $N = 100$ species (see Fig. 4 in the revised main text). This validation includes a detailed numerical analysis of the success rate of the proposed control approach with respect to: (1) changes in the ecological network parameters of the community (connectivity and inter-species interaction strength); (2) proportion of species that are used as driver species; and (3) errors in the ecological network used to identify the driver species and to design the linear Model Predictive Controller (quantified by a "rewiring probability"). Overall, the validation analyzed more than 7,700 in-silico microbial communities. The results of the numerical analysis shows that our proposed control framework achieves a high success rate (≥ 0.8), provided that the distance to the desired state is small, or the proportion of species that are driven is ≥ 0.06 (i.e., at least 6 from the 100 species are directly controlled).

Comments:

The first sentence of the abstract is a vague overstatement, while the last one is quite speculative.

We thank Reviewer #2 for this very constructive comment. In the revised abstract, we have improved the accuracy of the first sentence. We have also removed the last sentence of it, using the Discussion part of our manuscript to more explicitly discuss how our framework can be extended to control other biological systems (in particular, presenting an example of its application to control a biochemical oscillator).

The revised abstract reads:

Microbes comprise nearly half of all biomass on Earth. Almost every habitat on Earth is teeming with microbes, from hydrothermal vents to the human gastrointestinal tract. Those microbes form complex communities and play critical roles in maintaining the integrity of their environment or the well-being of their hosts. Controlling microbial communities can help us restore natural ecosystems and maintain healthy human microbiota. Yet, our ability to precisely manipulate microbial communities has been fundamentally impeded by the lack of a systematic framework to control them. Here we fill this gap by developing a

control framework based on the new notion of structural accessibility. This framework allows identifying minimal sets of “driver species” through which we can achieve feasible control of the entire microbial community. We numerically validate our control framework on large ecosystems, and then we demonstrate its application for controlling the gut microbiota of gnotobiotic mice infected with *Clostridium difficile* and the core microbiota of the sea sponge *Ircinia oros*.

Key microorganisms can be identified based on network topology in a structural sense (see <https://www.frontiersin.org/articles/10.3389/fmicb.2014.00219/full>, <https://www.nature.com/articles/srep15920>). To infer dynamical behaviour (not necessarily controlling, even predicting or only understanding) is a massive challenge...

We thank Reviewer #2 for this very constructive comment and for pointing out these two relevant references:

- Berry, David, and Stefanie Widder. "Deciphering microbial interactions and detecting keystone species with co-occurrence networks." *Frontiers in microbiology* 5 (2014): 219.
- Jordán, Ferenc, Mario Lauria, Marco Scotti, Thanh-Phuong Nguyen, Paurush Praveen, Melissa Morine, and Corrado Priami. "Diversity of key players in the microbial ecosystems of the human body." *Scientific reports* 5 (2015): 15920.

In the first paper, Berry and Widder characterized scenarios when co-occurrence networks provide a good proxy for the actual ecological (i.e., the interaction) network. The authors used this characterization to identify keystone species. In the second paper, the authors use the SpareCC method to calculate a correlation network from relative abundance data, leading to an undirected network. This network is then used to identify “important” species from a network perspective (as quantified by the degree and betweenness centrality) and from the perspective of which species appears more frequently.

Note that identifying the driver species (as defined in our main text) necessarily requires using the ecological network of the microbial community. In general, the ecological network and correlation-based networks can be very different (see, e.g., Fig 1 in Fisher, Charles K., and Pankaj Mehta. “Identifying keystone species in the human gut microbiome from metagenomic time series using sparse linear regression.” *PLOS One* 9.7 (2014): e102451). Thus, we consider the reference by Berry and Widder suggested by Reviewer #2 very illuminating and relevant to our present paper.

Considering Reviewer #2’s remarks, we have included the following paragraph to the main text (page 5):

...This network is defined as a directed graph where nodes $X = \{x_1, \dots, x_N\}$ represent species and edges $(x_j \rightarrow x_i) \in E$ denote that the j -th species has a direct ecological impact (e.g., direct promotion or inhibition) on the i -th species (Fig. 1a). Mapping these ecological networks requires performing mono-culture and co-culture experiments [20, 21], using time-resolved abundance data and system identification techniques [22, 23], or

using steady-state abundance data via a recently developed inference method [24]. The accuracy of all these methods strongly depends on how informative is the available data [25]. Note that these ecological networks are different from correlation or co-occurrence based networks because correlation doesn't imply causation [26]. Correlation-based networks can be readily constructed from abundance profiles of different samples [20, 27] and, under certain specific conditions [28], they could be a proxy of the underlying ecological network.

We hope the above paragraph clarifies the differences between ecological and correlation/co-occurrence networks. The work of Berry and Widder is cited as Ref. [28].

...I missed an overview (kind of metaanalysis) of earlier papers and the comparison of earlier results to the ones of this approach.

Finding "core" sets of organisms is a very nice idea and very promising (see <https://journals.aps.org/pre/abstract/10.1103/PhysRevE.65.026103>, <https://www.sciencedirect.com/science/article/pii/S1470160X1730359X>). This multi-species view may be based only on network structure or dynamics. To what extent the results can be generalizable here?

We thank Reviewer #2 for this very constructive comment and for the two very interesting references:

- Jain, Sanjay, and Sandeep Krishna. "Crashes, recoveries, and "core shifts" in a model of evolving networks." *Physical Review E* 65, no. 2 (2002): 026103.
- Ortiz, Marco, Brenda Hermosillo-Nuñez, Jorge González, Fabián Rodríguez-Zaragoza, Iván Gómez, and Ferenc Jordán. "Quantifying keystone species complexes: ecosystem-based conservation management in the King George Island (Antarctic Peninsula)." *Ecological Indicators* 81 (2017): 453-460.

The first one analyzes a mathematical model for a community where (molecular) species interact through a directed unweighted network. Repeatedly, those species with the lowest abundance are removed from the community, and new low abundant species are introduced by wiring them randomly to the existing species. The model produces "crashes and recoveries" in the number of species with nonzero abundance. The authors find these events are determined by changes in the so-called "autocatalytic sets" of the network (a subgraph such that each of its nodes has at least one incoming edge from another node in the same subgraph), which are determined by the topology of the network only. In the second paper, the authors calculate two "Keystone Species Complex" indices for an ecological community based on the sign of the ecological interactions and the abundance of species.

We emphasize that, in general, the driver species of a community are not the same as its keystone or "core" species. For example, keystone species are typically understood as species that have a disproportionate deleterious effect on the community upon their removal relative to their abundance. By contrast, the abundance of species does not directly influence which species are driver species, because the driver species are directly determined by the population dynamics of the community.

Considering Reviewer #2's remark, we have included the following paragraph to the Discussion part of the main text:

An influential method to understand and manage complex ecosystems has been identifying species with a "big impact" on the entire ecosystem, leading to notions such as keystone [45,46] or core [47] species. In general, the keystone or core species of an ecosystem are not necessarily its driver species. For example, the driver species of an ecosystem do not depend on their abundance, while the definition of keystone species does depend on the abundance —namely, species whose removal cause a disproportionate deleterious effect relative to their abundance [45].

The above paragraph clarifies how our approach compares to previous approaches in identifying "important" species, such as keystone and core species. We have also extended our bibliography to incorporate the above two references (Refs. [45] and [47]) suggested by Reviewer #2.

In an ecological (but also biological) context, the word "control" sounds very positivist. It is used for "biological control" (see https://link.springer.com/chapter/10.1007%2F1-4020-4767-3_7) but also these simplistic approaches bring lots of surprises. In multi-species, multi-parameter, highly non-linear ecodynamics, this seems to be not easily predictable by network structure. So, I disagree with the statement "The difficulty in this second challenge originates in our insufficient knowledge of the microbial dynamics and their interactions", I think this is inherent, not knowledge-limited.

We totally agree with this insightful remark of Reviewer #2: the difficulty of controlling a microbial community can also be due to the inherent nonlinear population dynamics that they may have. To clarify this point, we have revised the main text as follows (page 3):

...The difficulty in solving this second challenge is not only due to our insufficient knowledge of microbial dynamics and interactions, but also because of the inherently complex dynamics they often display.

The above sentence explicitly acknowledges the important role of the "multi-species, multi-parameter and highly nonlinear ecodynamics" in controlling microbial communities.

"disruption to the gut microbiota" and "driving ... back to their healthy stage" are to be discussed more carefully, given the extreme spatial, temporal and individual-level variability of these microbes and ALSO their systems. They are in continuous change and any "disruption" is much more like a change of some trajectory than "loosig a part" or something like this. This might be better discussed and presented.

We thank Reviewer #2 for this very insightful comment. We fully agree Reviewer #2 that modeling host-associated microbial communities, such as the human gut microbiota, is very challenging, due to the complex spatial structure of the gastrointestinal tract as well as host-microbe interactions. We emphasize that, when modeling a dynamical ecosystem, one must first decide how complex the model needs to be so as to capture the phenomenon of interest. A detailed model of the human microbiome would include mechanistic interactions among microbes, spatial structure of the particular body site, as well as host-microbiome interactions. Fortunately, in our

modeling we focus on exploring the impact that manipulating a subset of species (i.e., the driver species) has on the abundance of the other species. To achieve that, a population dynamics model written as a set of ordinary differential equations (ODEs) is sufficient. Indeed, this ODE-based modeling approach has been adopted in several previous studies in studying host-associated microbial communities, as illustrated by the following influential references:

- Katharine Z Coyte, Jonas Schluter, and Kevin R Foster. The ecology of the microbiome: Networks, competition, and stability. *Science*, 350(6261):663–666, 2015.
- Friedman, Jonathan, Logan M. Higgins, and Jeff Gore. "Community structure follows simple assembly rules in microbial microcosms." *Nature ecology & evolution* 1, no. 5 (2017): 0109.
- Faust, Karoline, and Jeroen Raes. "Microbial interactions: from networks to models." *Nature Reviews Microbiology* 10, no. 8 (2012): 538.
- Buffie, Charlie G., Vanni Bucci, Richard R. Stein, Peter T. McKenney, Lilan Ling, Asia Gobourne, Daniel No et al. "Precision microbiome reconstitution restores bile acid mediated resistance to *Clostridium difficile*." *Nature* 517, no. 7533 (2015): 205.
- Bucci, Vanni, Belinda Tzen, Ning Li, Matt Simmons, Takeshi Tanoue, Elijah Bogart, Luxue Deng et al. "MDSINE: Microbial Dynamical Systems INference Engine for microbiome time-series analyses." *Genome biology* 17, no. 1 (2016): 121.
- Panikov, N. S. "Understanding and prediction of soil microbial community dynamics under global change." *Applied Soil Ecology* 11, no. 2-3 (1999): 161-176.

Considering Reviewer #2 remark, we have included the following paragraph to the main text (Problem Statement, page 4):

In our modeling framework, we focus on exploring the impact that manipulating a subset of species has on the abundances of other species. We thus consider a microbial community whose *state* at time t can be determined from the abundance profile $x(t) \in \mathbb{R}^N$ of its N species, where the i -th entry $x_i(t)$ of $x(t)$ represents the abundance of the i -th species at time t ...

We have also revised Supplementary Note 1 to emphasize the assumptions we make as follows:

In our modeling framework, we focus on exploring the impact that manipulating a subset of species (i.e., the driver species) has on the abundances of other species. To achieve that, a population dynamics model written as a set of ODEs is sufficient. More precisely, we make the following assumption:

Assumption 1.

- a) The *state* of a microbial community can be determined by the abundance profile $x \in \mathbb{R}^N$ of its N species. Here, the i -th entry x_i of this vector is the absolute abundance of the i -th species.
- b) Spatial organization effects can be incorporated into the ecological network between the microbial species (e.g., determining if two species can interact or not).
- c) Host-microbe interactions and environmental factors remain constant during the time interval the control is performed.
- d) Stochastic effects, such as drift [10, 11], are negligible.

Additionally, to better discuss our assumptions, we added Remark 1 to the Supplementary Note 1, explaining how Assumption 1 translates into our modelling:

Remark 1.

- Assumptions 1a and 1b justify the use of a population dynamics model based on ODEs to describe the temporal evolution of the state of a microbial community.
- Due to Assumption 1c, host-microbe interactions and other environmental factors can be considered as constant parameters of such models.
- Assumption 1d allows us to use deterministic ODEs instead of stochastic ones, considerably simplifying the mathematical analysis.

"Note that this ecological network is fundamentally different from correlation or co-occurrence networks, because those networks are undirected." - there are differences from the viewpoint of symmetry and direction, this is clear, but I would not say "fundamentally". The association-based, statistical interaction networks are quite good proxies for "real" interaction networks. Most interactions are bidirectional (even if not symmetrical), so the effects can be considered undirected (if we do not consider the signs). The problem is that one cannot construct complex microbial networks without these proxies, see the methods mentioned just in the followings.

We thank Reviewer #2 for this comment. In the revised manuscript, we have revised this sentence as follows:

...This network is defined as a directed graph where nodes $X = \{x_1, \dots, x_N\}$ represent species and edges $(x_j \rightarrow x_i) \in E$ denote that the j -th species has a direct ecological impact (e.g., direct promotion or inhibition) on the i -th species (Fig. 1a). Mapping these ecological networks requires performing mono-culture and co-culture experiments [20, 21], using time-resolved abundance data and system identification techniques [22, 23], or using steady-state abundance data via a recently developed inference method [24]. The accuracy of all these methods strongly depends on how informative is the available data [25]. Note that these ecological networks are different from correlation or co-occurrence based networks because correlation doesn't imply causation [26]. Correlation-based networks can be readily constructed from abundance profiles of different samples [20,27] and, under certain specific conditions [28], they could be a proxy of the underlying ecological network.

"But that would be overkill" is unclear to me...

We thank Reviewer #2 for this comment. We have revised the paragraph containing that sentence as follows (page 7 of the main text):

Notice that when all species are directly controlled (i.e., each species is actuated by an independent control input so $M = N$ and $g(x)$ is full rank), the state of the whole microbial community can obviously be fully controlled. Fortunately, as we next show, controlling all the species in a community is far from being necessary. Indeed, several species can be *indirectly* controlled by the same control input when this signal is adequately propagated through the ecological network underlying the community.

In this revised paragraph, the word “overkill” was replaced by more detailed description of why it is unnecessary to directly control all the species in a community.

Figure 2b: the second control input is on x_1 , not on x_2

We thank the Reviewer pointing out this typo. It has been corrected in the revised manuscript.

Selecting driver species for control can be problematic since there is an interplay between network topology and actual abundance values. Highly abundant organisms can be popular partners (prey or other). This adds another level of complexity, I think. Can this be treated simply and efficiently?

We thank Reviewer #2 for this comment. Mathematically, the abundance of species is only **indirectly** related to which species are driver species. To see this point, note that the abundance of species is determined by the population dynamics of the community. In turn, the absence of autonomous elements—the condition that characterizes the driver species—is also determined by the same population dynamics of the community. Therefore, the abundance of species is only indirectly related to which species are driver species through the population dynamics of the community.

To clarify this point, we included the following sentence in the Discussion part of the main text:

..., the driver species of an ecosystem do not depend on their abundance, while the definition of keystone species does depend on the abundance—namely, species whose removal cause a disproportionate deleterious effect relative to their abundance [45].

Figure 4: we see 5 drivers for 9 non-drivers (mice) and 10 drivers for 10 non-drivers (spoge), I hope I counted well, and the non-drivers are typically perpheric, poorly connected nodes. So, almost all well-connected node needs to be directly controlled in order to indirectly control roughly the same number of poorly connected nodes - I am not sure this is technically feasible or plausible, given the hundreds of microbial species in realistic (really complex) communities.

We thank Reviewer #2 for this insightful comment. The ratio (minimal number of driver species)/(total number of species) will be determined by the topological properties of the ecological network of the community. For example, in general, the minimal number of driver species will decrease as the connectivity of the network increases (e.g., a fully connected network requires one driver species because there is a path that covers all species). Therefore, if the complexity of the community is measured by the number of interactions between its species, then **more complex communities require a smaller number of driver species**.

In addition, we emphasize there is a tradeoff between the chosen set of driver species and how complicated is to design the control signal that should be applied to driver the microbial community toward a desired state. If a set of N driver species is used (i.e., all species are directly driven), then the control signal takes its simplest form because it is directly possible to set the abundance of each species to the desired value. As the size of the set of driver species decreases, we expect that the control signal needed to steer the microbial community to the desired state is more complicated.

Considering the above two points, we have added the following sentence to the Discussion part of our revised manuscript:

Note that, in general, it can be expected that the design of control strategies becomes more difficult as the number of used driver species decreases (see Remark 9 in Supplementary Note 5). Additionally, we note that despite the minimal number of driver species decreases as the ecological network becomes denser, this condition is only sufficient. Indeed, the minimal number of driver species of a microbial community should be mainly determined by the degree distribution of the ecological network, since the maximum matching size of a directed network is largely determined by its degree distribution [51].

Finally, we have included the following remark to the revised Supplementary Note 5:

Remark 9. There exists a tradeoff between the number of driver species and the complexity of the control signal needed to drive the microbial community towards the desired state. To illustrate this point, let's consider the microbial community of Example 6 with $N = 3$ species and $M = 2$ driver species. Here it is necessary to use the two impulsive control inputs of Eq. (S16) to reach the desired state. The component u_{10} of these control inputs is rather complicated as shown in Eq. (S15). By contrast, if we enlarge the set of driver species to $M = 3$ (i.e., we control all species), then a single control input $u(t_0) = x_d - x(t)$ can evidently drive the microbial community to x_d . Note also that the control signal is simpler compared to the case of using $M = 2$ driver species. In general, it can be expected that the complexity of the signal needed to drive the community increases as the number of driver species decrease.

“we found that the conditions for the absence of autonomous elements for the continuous and the impulsive control schemes are identical” - differences probably emerge if more complicated functional response kinetics are considered with saturation terms and spatial constraints.

We thank Reviewer #2 for this insightful comment. Mathematically, the fact that the conditions of Theorems 1 and 2 are identical (see Supplementary Note 1) imply that the conditions for the absence of autonomous elements in a pair $\{f, g\}$ are identical for continuous and impulsive control inputs. Nevertheless, we agree with Reviewer #2 that the conditions for the absence of autonomous elements for continuous and impulsive control actions can be different when more detailed models of those control actions are used. This is because, under more detailed models, different control actions could lead to different pairs $\{f, g\}$.

Considering Reviewer #2's comment, we have revised the following paragraph in the Discussion part of the main text (page 22):

In practice, the performance of the control algorithms can also be improved by using more detailed models that incorporate the dynamics of the susceptibility of species to the control actions (e.g., the pharmacokinetics of prebiotics). In such case, different control actions could be modeled by different pairs $\{f, g\}$ in Eqs. (2) or (3), making the conditions for the absence of autonomous elements different for continuous and impulsive control actions.

"Note that once G_c is structurally accessible, this condition will not be violated if new edges are added to the network" - this is an important finding, but how sensitive is the identification of drivers to losing edges?

We thank Reviewer #2 for this very constructive comment. In the revised main text, we have included a new analysis of the sensitivity of our control framework to errors in the ecological network (see Fig. 4h). Specifically, we present a numerical analysis of the “success rate” of our control framework for communities of $N = 100$ species against 5%, 10% and 15% errors in the ecological network that was used to identify the driver species and to calculate the linear MPC. We find that our framework is robust to such errors, with a 5% error in the network leading to a drop of about 0.3% in its success rate. This result emphasizes the importance of having an accurate network reconstruction in order to achieve a high success rate in controlling microbial communities.

I think it is a major question how can controllability depends on the two states (initial and desired). If the initial state would naturally evolve towards the desired one, or just in the contrary direction, might be crucial here. So, instead of looking at trajectories between two points, it would be nice to better discuss the positions of these two points in the landscape.

We thank Reviewer #2 for this comment. *First*, as Reviewer #2 remarks, the controllability of nonlinear systems —defined as the ability to steer the system from an initial to a desired state— depends on both the initial state and the desire state. Checking if a nonlinear system is controllable or not is an NP-hard problem, even for systems with mild nonlinearities (e.g., bi-linear systems). This is discussed, e.g., in the following reference: Sontag, Eduardo. "From linear to nonlinear: some complexity comparisons." *Decision and Control, 1995., Proceedings of the 34th IEEE Conference on*. Vol. 3. IEEE, 1995. A consequence of the computational intractability of the controllability of nonlinear systems is that, in general, it is infeasible to analyze the “controllability landscape” between all pairs of states, as Reviewer #2 suggests. We emphasize that the notion of accessibility (i.e., the absence of autonomous elements) used in our paper was introduced by the control theory community as an alternative to circumvent the infeasibility of checking the controllability of nonlinear systems. This notion has been instrumental in the development of nonlinear control theory, and it is a keystone behind technological advances such as robotics (see, e.g., Ref. [30] of the main text).

Second, even if it was feasible to check the controllability of nonlinear systems, it would be impossible to test the controllability of most microbial communities because we lack detailed knowledge of their population dynamics. In other words, constructing the “controllability landscape” would be impossible because we lack the necessary detailed knowledge of its population dynamics.

Finally, regarding Reviewer #2’s remark that “...If the initial state would naturally evolve towards the desired one, or just in the contrary direction, might be crucial here...”, we point out that if for an initial state the microbial community naturally evolves towards the desired state, then it is not necessary to control the community at all. This would correspond, for example, to a dysbiotic gut microbiota that spontaneously recover without any external intervention (i.e., without control

actions). In our framework, we consider that control is necessary because the state of the microbial community will not naturally evolve to the desired state.

Considering Reviewer #2's remark, we revised the main text to clarify that control is needed because the community will not naturally evolve to the desired state (page 5 of the main text):

Controlling a microbial community consists in driving its state from an initial value $x_0 \in \mathbb{R}^N$ at time $t = 0$ (e.g., a “diseased” state) towards a desired value $x_d \in \mathbb{R}^N$ (e.g., a “healthier” state, Fig. 1b). We consider that the community will not naturally evolve to the desired state. To drive the microbial community, we consider a set of M control inputs $u(t) \in \mathbb{R}^M$ that directly affect...

The Discussion part of the revised main text also contains the following paragraph to clarify the advantages of the notion of accessibility with respect to the notion of (nonlinear) controllability (page 21):

It was suggested before that notion of *controllability*—the ability to drive a system between any two states— could help predicting the success of ecosystem management strategies [48]. For microbial communities and many other biological systems, it is inadequate to use the notion of controllability because there are state that those systems cannot reach by their nature (e.g., those states corresponding to negative abundances). Additionally, since dynamic models for microbial communities and other complex ecosystems are nonlinear, uncertain, and often very difficult to infer, it is impossible to even test if those systems are controllable or not. The notion of structural accessibility at the basis of our framework overcomes these two limitations, generalizing the control-theoretic notion of accessibility [32] to systems with uncertain dynamics and impulsive control inputs...

Finally, we thank again Reviewer #2 for her/his very insightful and constructive comments. We hope our responses above have addresses those very legitimate issues/concerns in a satisfactory manner.

Response to Reviewer #3

In ms NCOMMS-18-15520, the authors present a powerful method for controlling the dynamics of microbial communities, given relatively little information about the nature of their interactions. Importantly, the methodology is applicable to cases where the dynamics are nonlinear, in contrast the "structural controllability" which is currently a very hot topic in network science.

The work appears to be without technical error and the manuscript is mostly clear, though the work is technically dense and will likely be challenging for some readers who would otherwise be interested in the work. My specific comments, below, are mostly aimed toward increasing the readability of the work (see especially comment 2, where I encourage the inclusion of more detail in the main text).

I hope these comments are useful to the authors as they revise their ms.

We thank Reviewer #3 for her/his very positive assessment of our work. Below we provide a point-by-point response to Reviewer #3's comments.

Comment 1. It seems to me that the two problems mentioned at the top of page 3 (knowing the set of minimal species necessary for control and then knowing what that control is) are co-implicated to some degree. It may be prudent for the authors to make this point clear.

We thank Reviewer #3 for this insightful comment. Certainly, there is a tradeoff between the chosen set of driver species and how complicated is to design the control signal that should be applied to driver the microbial community toward a desired state. If a set of N driver species is used (i.e., all species are directly driven), then the control signal takes its simplest form because it is directly possible to set the abundance of each species as desired. As the size of the set of driver species decreases, we expect that the control signal needed to steer the microbial community to the desired state become more complicated.

Considering Reviewer #3 comment, we have added the following sentence to the Discussion part of our revised manuscript:

Note that, in general, it can be expected that the design of control strategies becomes more difficult as the number of used driver species decrease (see Remark 9 in Supplementary Note 5).

Additionally, we have included the following remark to the revised Supplementary Note 5:

Remark 9. There exists a tradeoff between the number of driver species and the complexity of the control signal needed to drive the microbial community towards the desired state. To illustrate this point, consider the microbial community of Example 6 with $N = 3$ species and $M = 2$ driver species. Here it is necessary to use the two impulsive control inputs of Eq. (S16) to reach the desired state. The component u_{10} of these control inputs is rather complicated as shown in Eq. (S15). By contrast, if we enlarge the set of driver species to $M = 3$ (i.e., we control all species), then a single control input $u(t_0) = x_d - x(t)$ can evidently drive the microbial community to x_d . Note also that the control

signal is simpler compared to the case of using $M = 2$ driver species. In general, it can be expected that the complexity of the signal needed to drive the community increases as the number of driver species decrease.

Comment 2. Near the bottom of page 7, the authors define the term "autonomous element" and provide an example (Figure 2) explaining the term. However, the authors do not provide an explanation in terms of the topology of the network for why it is that node x_3 cannot (by itself) be a driver species, and instead refer the reader to a number of supplementary notes.

It would be very useful if the authors could provide an intuitive explanation for why it is that controlling node x_3 confines the state of the network to a low-dimension manifold, then refer the reader to the supplementary notes for additional details.

We thank Reviewer #3 for this very legitimate concern. We have revised the main text to improve the explanation of why Fig. 2a contains an autonomous element as follows:

...More precisely, our mathematical formalism reveals that $\xi = x_1x_2$ is an autonomous element for this microbial community (Example 2 in Supplementary Note 2). Indeed, differentiating ξ with respect to time yields $\dot{\xi} = x_1x_2(1 - x_3) + x_1x_2(-1 + x_3) \equiv 0$, which implies that the state of the community is constrained to the low-dimensional manifold $\{x \in \mathbb{R}^3 \mid x_1x_2 = x_1(0)x_2(0)\}$ for all control inputs (Fig. 2a right). Intuitively, an autonomous element exists because the control input cannot change the abundance of species x_1 without changing the abundance of species x_2 in a predefined way (i.e., $x_2 = x_1(0)x_2(0)/x_1$).

The above text contains an intuitive explanation of why the autonomous element exists, as suggested by Reviewer #3.

Comment 3. Typo, line 11 of page 11 ('\alpha' -> j, if I read the authors correctly)

We have corrected this typo in the revised manuscript. Thanks!

Comment 4. More justification/explanation should be given for the choice of A^\wedge given on line 14 of page 17.

We thank the Reviewer for this legitimate concern. In the revised manuscript, we have improved our description for the choice of \hat{A} as follows:

Based on the ecological network of this community and its population dynamics (see Fig. 1 and its caption), we choose $\hat{A} = (-0.5, 0, -0.1; 0, -5, 1; 0, 0, -1)$ as a proxy for its interaction matrix. Note that \hat{A} is a rather rough approximation of the linearization of the population dynamics at the desired state given by $(-0.37, 0, -0.05; 0, -5.31, 0.52; 0, 0, -1)$.

Comment 5. The authors may be interested in some broadly related work that also examines the topology of interactions networks to characterize and/or control network dynamics. I stress the authors needn't feel compelled to cite either of these works:

DOI: 10.1063/1.4809777

DOI: 10.1186/1752-0509-8-53

We thank Reviewer #3 for pointing out these two very interesting works, which have been cited in the revised manuscript.

Finally, we would like to thank Reviewer #3 again for her/his very insightful and constructive comments. We hope our responses above have addresses those very legitimate issues/concerns in a satisfactory manner.

Response to Reviewer #4

Marco Tulio Angulo and colleagues describe in their manuscript ‘Controlling complex microbial communities: a network based approach’ their strategy on how to access microbial communities through precise manipulation. To do so the authors describe their approach of a control framework based on the notion of structural accessibility. Through this approach they claim their ability to identify ‘driver species’ through which manipulation becomes feasible.

General comments:

I very much like the authors concept since their method is practical, straight forward and a tangible application of network analysis which has remained too descriptive in many previous studies...

We thank Reviewer #4 for her/his very effective summary of our paper and for her/his time reviewing our paper. We are also glad that she/he finds the results in our paper of potential interest.

...Despite the great potential of the described method, experiments for validation provided by the authors are only run in simulations and only for a limited number of nodes. Of course it would have been amazing if the authors could have proven their hypothesis through real control experiments: for example by manually increasing / decreasing *Clostridium difficile* abundance in a lab experiment. Alternatively there are data-sets of community disturbance available in the literature. It would have been great to see if the MPC model correctly predicts the network rewiring.

We appreciate this constructive comment. We agree with Reviewer #4 that it is extremely relevant to obtain an experimental validation of our control method. Yet, as history has shown for controlling other bio/technological systems such as robots, aircraft and biological reactors, we believe that the feasible and efficient control of complex microbial communities will be achieved only after the necessary theory has been developed. In this sense, we consider that those experimental results are out of the scope of our present paper that focus on providing the theoretical basis.

Datasets of community disturbances available in the literature cannot be readily leveraged to validate our framework either, because validation of our framework would require performing very special sequential interventions to a microbial community, such as the sequence of increase/decrease in the abundance of the identified driver species in Fig. 3a. It is very unlikely that such specific sequence of interventions was done in any available experimental dataset. Hence, we don't expect that any existing dataset could provide evidence against or in favor of the effectivity of our control approach.

As mentioned above, I find the paper sound and novel. The authors, however, are sparse on highlighting the novelty of their approach in the paper. It would be good to highlight in more detail the novelty compared to available approaches. I see the novelty in the accessibility and controllability of the approach.

We are glad that Reviewer #4 finds our results sound and novel. We also apologize for not clearly highlighting the novelty of our approach. As Reviewer #4 effectively summarizes, the novelty of our results include:

- 1) We introduce a new definition of *autonomous element* for impulsive control systems (Definition 3 in Supplementary Note 2), which has never been defined in the literature.
- 2) We characterize both necessary and sufficient conditions for the absence of autonomous elements in arbitrary nonlinear impulsive control systems (Theorem 2 in Supplementary Note 2). This can be used to better model certain control actions applied to microbial communities.
- 3) We introduce the notion of *structural accessibility* as: (i) a generalization of the notion of accessibility to uncertain nonlinear systems; and (ii) as a generalization of the notion of (linear) structural controllability to nonlinear systems. To define this notion, we also introduce the notions of a *base model* (the “simplest” dynamics that the microbial community may have, chosen as the Generalized Lotka-Volterra model) and their deformations (i.e., the more “complex” dynamics that the community may have).
- 4) We provide a complete *graph theoretical* or *network characterization* of structural accessibility (Theorem 3 in Supplementary Note 3). This characterization relies on proving that increasing the complexity of a deformation cannot (generically) invalidate a set of driver species (Proposition 1 in Supplementary Note 3).
- 5) Based on the above characterization of structural accessibility, we introduce a method to identify a minimal set of driver species of a microbial community solely from the topology of its ecological network (Supplementary Note 4).
- 6) When the population dynamics of the community is known, we construct an impulsive nonlinear Model Predictive Controller (MPC) to generate the control signal that needs to be applied to the driver species to steer the community towards a desired state (see Eq. (6) and Fig. 3a,b of the main text).
- 7) To actually implement the above impulsive nonlinear MPC controller, we built a mathematically rigorous method to calculate a sufficient number of impulses needed to drive the community towards the desired state (Theorem 4 in Supplementary Note 4).
- 8) When no detailed knowledge of the population dynamics of the community is available, we proposed a linear MPC based on a proxy of the interaction matrix and susceptibility matrix of the community. We demonstrate that this approach can succeed in steering microbial communities towards desired states, despite of the uncertain nonlinear population dynamics (Fig. 3c,d). In the revised main text, we include a detailed validation of the linear MPC applied to the driver species for controlling large microbial communities ($N = 100$ species) with nonlinear population dynamics and diverse ecological network parameters (Fig. 4).
- 9) We illustrate by simulation the application of our framework in controlling the gut microbiota of mice infected with *C. difficile*, and the core microbiota of the sea sponge *Ircinia oros*, without using detailed knowledge of their population dynamics.
- 10) We illustrate the potential application of our framework to control other biological systems, e.g., the Repressilator (a synthetic genetic regulatory network, see Supplementary Figure 1).

Taken together, we believe the above contributions represent significant advances in controlling complex networked systems by introducing and characterizing the notion of structural accessibility

for nonlinear systems with impulsive or continuous control. In particular, we demonstrated the application of our control theoretical framework in the field of controlling microbial communities, by providing a systematic pipeline to efficiently steer large microbial communities towards desired states.

To address this issue, we have carefully revised the Introduction and Discussion parts of our paper to better highlight the novelty of our approach. In particular, the following paragraph was added to the Introduction part:

Structural accessibility is a generalization of the notion of structural controllability [16] — which only applies to systems with linear dynamics— to systems with nonlinear dynamics. Linear structural controllability is receiving increasing attention from the viewpoint of Network Science [17].

The above paragraph highlights the novelty of the notion of structural accessibility we propose in the Network Science context, as Reviewer #4 suggested. Additionally, we included the following paragraph in the Discussion part of the main text:

It was suggested before that notion of *controllability* —the ability to drive a system between any two states— could help predicting the success of ecosystem management strategies [48]. For microbial communities and many other biological systems, it is inadequate to use the notion of controllability because there are states that those systems cannot reach by their nature (e.g., those states corresponding to negative abundances). Additionally, since dynamic models for microbial communities and other complex ecosystems are nonlinear, uncertain, and often very difficult to infer, it is impossible to even test if those systems are controllable or not. The notion of structural accessibility at the basis of our framework overcomes these two limitations, generalizing the control-theoretic notion of accessibility [32] to systems with uncertain dynamics and impulsive control inputs. As result, our framework allows efficiently controlling microbial communities only knowing their underlying ecological networks...

The above paragraph highlights the novelty of our approach from the viewpoints of Control Theory and Ecology.

The main claim of the paper that is to develop a reliable method to identify driver nodes for manipulation of large networks has been confirmed and proven in theory very elegantly, the paper, however, lacks experimental confirmation which could be by using available datasets. Unfortunately simulations do not take into consideration large networks although the authors write about large networks. Networks from natural samples (or ecological networks) contain rarely below 100 nodes and therefore I would not consider a network with < 100 nodes a large network.

We thank Reviewer #4 for rising this very legitimate concern. To address this issue, in Fig. 4 of the revised main text we provide a new detailed in-silico validation of our control framework for large ecosystems with 100 species. This result shows that our control framework can successfully drive large microbial communities using their driver species despite uncertainty on its population dynamics.

Concrete suggestions:

1) The authors should certainly re-think the term and usage of ‘ecological networks’. They could just tone down the text and re-write those parts.

We thank Review #4 for this comment. In the revised manuscript, we offer a clearer definition of ecological networks as follows (see page 5 of the main text):

...This network is defined as a directed graph where nodes $X = \{x_1, \dots, x_N\}$ represent species and edges $(x_j \rightarrow x_i) \in E$ denote that the j -th species has a direct ecological impact (e.g., direct promotion or inhibition) on the i -th species (Fig. 1a). Mapping these ecological networks requires performing mono-culture and co-culture experiments [20, 21], using time-resolved abundance data and system identification techniques [22, 23], or using steady-state abundance data via a recently developed inference method [24]. The accuracy of all these methods strongly depends on how informative is the available data [25]. Note that these ecological networks are different from correlation or co-occurrence based networks because correlation doesn’t imply causation [26]. Correlation-based networks can be readily constructed from abundance profiles of different samples [20,27] and, under certain specific conditions [28], they could be a proxy of the underlying ecological network.

Hopefully, this term will not cause confusions any more.

2) The authors should certainly check available data-sets and use ‘real’ data examples to show that the model is trained and deployed correctly.

We thank Reviewer #4 for this very constructive comment. We did check available sets and could not find a suitable real dataset that can provide evidence against or in favor of the effectivity of our control approach. In the current work, we focus on building the theoretical foundation of the realistic control of complex microbial communities. We consider that a theory could advance and guide the design of experiments, which could be a future work.

Some more detailed comments:

The main criticism is that the authors take a number of assumptions into account which can be hardly justified:

A) The Assumption that the starting network is directed: in very few cases in ecology or in microbiology there is enough data available to infer a directed network. With metagenomics data even more. I therefore challenge this assumption.

We thank Reviewer #4 for pointing out this very legitimate concern. We completely agree with Reviewer #4 that inferring the directed ecological underlying a microbial community is in general a very challenging problem. Indeed, many factors make the network inference problem very difficult, from intrinsic limitations due to the available data (e.g., only steady-state samples of the human gut microbiota are available), to a large uncertainty about the population dynamics of the community.

On one hand, to address the above challenges, our group has recently developed a novel network inference method that uses steady-state samples and does not require any knowledge of the population dynamics of the microbial community: *Xiao, Yandong, Marco Tulio Angulo, Jonathan Friedman, Matthew K. Waldor, Scott T. Weiss, and Yang-Yu Liu. "Mapping the ecological networks of microbial communities." Nature communications 8, no. 1 (2017): 2042.* We validated our inference method in-silico (finding that roughly $5N$ samples are sufficient to accurately infer the network underlying a community of N species) and three experimental datasets. In particular, for experimental data, we find our method can correctly infer both the sign and direction of roughly 78% of all interactions. Therefore, we believe that an accurate map of the ecological networks of large complex microbial communities such as the human gut microbiota will become available in the coming years.

On the other hand, an important theoretical result of our paper is the observation that adding edges to an ecological network cannot invalidate a given set of driver species. This result means that the driver species of a community can be identified using an ecological network that includes only those interactions that were inferred with high confidence. Later, when more data is available and other interactions have high confidence, the set of driver species can be refined. In this sense, the driver species of a community can be identified even if the used ecological network is incomplete.

Considering Reviewer #4's remark, we have added the following paragraph to the revised main text (see page 5):

... Mapping these ecological networks requires performing mono-culture and co-culture experiments [20, 21], using time-resolved abundance data and system identification techniques [22, 23], or using steady-state abundance data via a recently developed inference method [24]. The accuracy of all these methods strongly depends on how informative is the available data [25]. Note that these ecological networks are different from correlation or co-occurrence based networks because correlation doesn't imply causation [26]. Correlation-based networks can be readily constructed from abundance profiles of different samples [20,27] and, under certain specific conditions [28], they could be a proxy of the underlying ecological network.

The above paragraph explicitly acknowledges that inferring ecological networks remains a challenging problem, but that the conjunction of better metagenomics technology and several recently developed methods offer promising results for inferring the ecological networks underlying complex microbial communities.

Additionally, we also revised the section "Identifying minimal sets of driver species in microbial communities" to clarify why it is possible to identify the driver species from an incomplete ecological network of the community:

...Note that once G^c is structurally accessible this network cannot lose its structural accessibility when new edges are added to it. This observation implies that a set of driver species remains valid even if new edges (e.g., new inter/intra-species interactions) are added to the ecological network of the community. Therefore, it is possible to find the driver species of a microbial community using an "incomplete" ecological network that only includes some of the ecological interactions (e.g., high-confidence interactions).

B) Driver species are definitely the way to go to gather more insights in network analysis. However, for intrinsic properties of networks, driver nodes correlate positively with the heterogeneity and density of the network. This means that identifying the driver nodes is relevant only if the network is dense enough. The authors do not make any assumption on network homogeneity and density. On the contrary at pg 19 line 4 they state that they found 10 driver nodes out of 20. This sounds like not a really convenient strategy in this case.

We thank Reviewer #4 for this very insightful comment. We agree with Reviewer #4 that a dense ecological network is a sufficient condition for having a small minimal number of driver species. However, this condition is far from being necessary. As an example, consider a large community with $N \gg 1$ species whose network is a directed path from species 1 to species N . Despite the density of this network is very low (i.e., actually $1/N$), the minimal set of driver species contains only one species (i.e., species 1 is a solo driver species). In this sense, the identification of driver species can be useful even if the ecological network of the community is not dense.

In general, we expect that the minimal number of driver species of a microbial community strongly depends on the degree distribution of its ecological network. This is because we have mapped the driver species identification problem to solving a maximum matching problem on directed networks (Proposition 3 in Supplementary Note 4). And, as shown in this paper (*Liu, Yang-Yu, Jean-Jacques Slotine, and Albert-László Barabási. "Controllability of complex networks." Nature 473, no. 7346 (2011): 167*), for a directed network, its maximum matching size is largely determined by its degree distribution.

Considering Reviewer #4's comment, we have included the following paragraph in the Discussion part of the revised main text:

Additionally, we note that despite the minimal number of driver species decreases as the ecological network becomes denser, this condition is only sufficient. Indeed, the minimal number of driver species of a microbial community should be mainly determined by the degree distribution of the ecological network, since the maximum matching size of a directed network is largely determined by its degree distribution [51].

Finally, we thank again Reviewer #4 for her/his very insightful and constructive comments. We hope our responses above have addresses those very legitimate issues/concerns in a satisfactory manner.

Reviewer #2 (Remarks to the Author):

Dear Authors, thank you for the great job you made during revision. My concerns have been appropriately answered. Especially Figure 4 will be great help for the Readers, I think.

Reviewer #3 (Remarks to the Author):

Dear Editor,

In their extensive revisions to ms NCOMMS-18-15520A, the authors have satisfactorily addressed the concerns raised in my initial review.

Reviewer #4 (Remarks to the Author):

The authors put significant effort in revising the manuscript and I am very happy on how they re-wrote the text. There are only some minor concerns left:

- 1) I can see that doing experiments to validate the proposed concept are out of scope of this manuscript. It is however hard to understand that there are no data sets available that would enable a test of the findings. This expectancy might come from the title and might misguide future readers as well. There should be somewhere written that this is a theoretical framework. For example: "Inferring controllability of complex microbial communities: a network-based approach".
- 2) The authors put significant effort into explaining how they use the term ecological networks. Probably due to my background I do not think just looking into microbe-microbe interactions justify the term "ecological" here, but as highlighted in the response this was published before. I think the discussion is already out so and in the present framework given in this manuscript environmental factors could be easily implemented by future users. A sentence in the discussion on future implementations could certainly clarify this.

REVIEWERS' COMMENTS:

Reviewer #1 (Remarks to the Author):

Report on NCOMMS-18-15520A

“Controlling complex microbial communities: a network-based approach”

General comments. I appreciate authors’ tremendous efforts in responding to the referee comments and in revising the manuscript. Having said this, I am still of the opinion that the manuscript adds little to our knowledge about controlling nonlinear dynamical networks. In spite of authors’ arguments to portrait their method as one for nonlinear networks, it is fundamentally a linear control method. Basically, from data, the authors construct a linear dynamical network to model the network system of microbial communities. As for any natural system in the real world, microbial communities are intrinsically nonlinear. To model the system using a linear network is not meaningful, let alone controlling the actual system using the linear model. The authors made very strong rebuttal and claimed that their control method works for nonlinear dynamical networks, but this is simply false.

The present work seems to be a further step in the relentless push of linear network control, a field that the senior author helped initiate. The natural world is nonlinear. Complex networks in the real world are nonlinear and their control requires **real** nonlinear method. To claim that a globally linear model can be used to control a nonlinear network is misleading. This practice will have negative impacts on the development of the field and needs to be stopped.

Technical comments. The idea of controlling a nonlinear network based on **locally** linearized dynamics and optimal cost function has been published [1]. In their response, the authors stated that “*We emphasize that in our method the pair (\hat{A}, \hat{B}) does not need to be any linearization of the population dynamics*”. This indicates that the authors did not even care that the actual network system is nonlinear and just tried to brutally fit the nonlinear system with a linear model from data - the so-called linear model predictive control (MPC). Especially, they used a proxy interaction matrix associated with the fitted linear networked dynamical system to calculate the required control signals for the nonlinear networked dynamical system. This procedure is not only misleading and meaningless but also fundamentally wrong.

It is not possible that, **globally**, one can approximate a nonlinear networked dynamical system by a linear model with some fitted interaction matrix $\{\hat{A}, \hat{B}\}$. Any set of control signals calculated from the linear model will in general have no relevance to the actual system. The linear approximation is valid but only **locally**. For a nonlinear networked system, locally the linearized systems are different from one point to another in the phase space. The control signals from authors’ MPC approach are irrelevant to the actual underlying nonlinear system.

As an example, the authors stated on page 19 in the main text that the proxy interaction matrix is $\hat{A} = (-0.5, 0, -0.1; 0, -5, 1; 0, 0, 1)$ and the linearization of the population dynamics at the desired state is $\hat{A} = (-0.37, 0, -0.05; 0, -5.31, 0.52; 0, 0, -1)$. The proxy interaction matrix is not even able to model the locally linear dynamics in the vicinity of the desired state (note the difference in the sign of the last element). How can one expect the proxy matrix to hold globally?

References

- [1] Isaac Klickstein, Afroza Shirin, and Francesco Sorrentino. Locally optimal control of complex networks. *Phys. Rev. Lett.*, 119(26):268301, 2017.

Response to Reviewer #1

General comments. I appreciate authors' tremendous efforts in responding to the referee comments and in revising the manuscript. Having said this, I am still of the opinion that the manuscript adds little to our knowledge about controlling nonlinear dynamical networks. In spite of authors' arguments to portrait their method as one for nonlinear networks, it is fundamentally a linear control method. Basically, from data, the authors construct a linear dynamical network to model the network system of microbial communities...

We thank Reviewer #1 for her/his time reviewing our paper. Having said this, we have to respectfully disagree with her/his two claims: 1) that our framework “is fundamentally a linear control method”, and 2) that “Basically, from data, the authors construct a linear dynamical network to model the network system of microbial communities”.

Regarding the first claim, we emphasize that for identifying the driver species of a community, our framework uses the notion of accessibility (i.e., the absence of autonomous elements). Our framework is not “fundamentally linear” simply because accessibility is not equivalent to linear controllability. We made this same point in the previous Response Letter, using the following simple example to illustrate a system that is accessible but not linearly controllable:

$$\dot{x}_1 = x_2^3, \quad \dot{x}_2 = u.$$

Consequently, we find that Reviewer #1's claim that our framework is “fundamentally linear” is not informed by any reason but by incorrect beliefs.

Regarding the second claim, we emphasize that our framework does not require constructing a “linear dynamical network” at all. Perhaps, Reviewer #1 missed the fact that Eqs. (4-6) of the main text handle nonlinear dynamics. As a simple and direct way to demonstrate that his/her second claim is incorrect, we invite Reviewer #1 to check the microbial community shown in Figure 1. This toy community has the following nonlinear population dynamics (see the legend of Fig. 1):

$$\begin{aligned}\dot{x}_1 &= 0.1 + x_1 \left(1 - \frac{x_1}{5}\right) \left(\frac{x_1}{3} - 1\right) - \frac{0.1x_1x_3}{1 + x_3}, \\ \dot{x}_2 &= 0.1 + x_2 \left(1 - \frac{x_2}{4}\right) (x_2 - 1) + \frac{x_2x_3}{1 + x_3}, \\ \dot{x}_3 &= x_3 \left(1 - \frac{x_3}{2}\right) (x_3 - 1).\end{aligned}$$

Our framework identifies x_3 as a solo driver species for this community. We emphasize this conclusion is not based on any linearization of the population dynamics (see Example 1 in Supplementary Note 2). Next, Figures 3a-b show that the nonlinear Model Predictive Control (MPC) of Eq. (6) provides the control signal that drives this community to the desired state. Using this nonlinear MPC, no “linear dynamical network” was built. Consequently, this example completely refutes Reviewer #1's claim that “the authors construct a linear dynamical network to model the network system of microbial communities.”

Finally, only when no detailed knowledge of the population dynamics of the microbial community is available, our framework proposes building a linear model to predict its response. Here, a robust control signal is built using a linear predictive model, leading to the proposed linear MPC. Note that this control signal will be applied to drive a community with nonlinear dynamics. The robustness of the linear MPC allowed us to control large complex communities with nonlinear dynamics, as shown in the systematic validation of Fig. 4 of the main text.

...As for any natural system in the real world, microbial communities are intrinsically nonlinear. To model the system using a linear network is not meaningful, let alone controlling the actual system using the linear model. The authors made very strong rebuttal and claimed that their control method works for nonlinear dynamical networks, but this is simply false.

We agree with Reviewer #1 that most natural systems are nonlinear. Not only that, the vast majority of man-made systems, from induction motors to airplanes, are also nonlinear. However, Reviewer #1 incorrectly concluded that for controlling a nonlinear system it is necessary to use a nonlinear controller. The control community did not make this same mistake. By contrast, they dedicated huge efforts to develop robust linear controllers that can cope, to certain extent, with the nonlinearities of the systems they are controlling. A successful example of this approach, applied to highly nonlinear systems such as aircraft, is the H_∞ robust optimal control (see, e.g., the classical reference Zhou, K., Doyle, J.C. and Glover, K., 1996. *Robust and optimal control*, vol. 40, p. 146., New Jersey: Prentice hall).

The present work seems to be a further step in the relentless push of linear network control, a field that the senior author helped initiate. The natural world is nonlinear. Complex networks in the real world are nonlinear and their control requires real nonlinear method. To claim that a globally linear model can be used to control a nonlinear network is misleading. This practice will have negative impacts on the development of the field and needs to be stopped.

We believe our contributions represent significant advances in controlling complex networked systems by introducing and characterizing the notion of **structural accessibility** for nonlinear systems with impulsive or continuous control. Moreover, we demonstrated the application of our control theoretical framework in the field of controlling microbial communities by providing a systematic pipeline to efficiently drive large microbial communities towards desired states. We consider this practice will have a huge impact on the further development of network control, and hence should be encouraged, rather than be stopped.

Technical comments. The idea of controlling a nonlinear network based on locally linearized dynamics and optimal cost function has been published [1]. In their response, the authors stated that “We emphasize that in our method the pair (\hat{A}, \hat{B}) does not need to be any linearization of the population dynamics”. This indicates that the authors did not even care that the actual network system is nonlinear and just tried to brutally fit the nonlinear system with a linear model from data - the so-called linear model predictive control (MPC). Especially, they used a proxy interaction matrix associated with the fitted linear networked dynamical system to calculate the required control signals for the nonlinear networked dynamical system. This procedure is not only misleading and meaningless but also fundamentally wrong.

Again, we find that Reviewer #1's opinion is informed by incorrect beliefs. The procedure to which Reviewer #1 refers as "...not only misleading and meaningless but also fundamentally wrong" is nothing but using the robustness of feedback controllers, the most fundamental property of feedback control systems.

In undergraduate courses of feedback control theory, a key lecture consists in explaining the need for robust controllers. Namely, because any mathematical model of a system is only an approximation of it, a controller must be robust in the sense that it should work despite a mismatch between the model used to design it and the true system dynamics. Otherwise, if a controller is not robust, it simply won't work in practice. Realizing this fact, the control community has placed a special emphasis on designing controllers that are "as robust as possible", in particular optimal robust linear controllers. For further details, we refer Reviewer #1 to classical books such as "Kemin Zhou and John Comstock Doyle. *Essentials of robust control*. Vol. 104. Upper Saddle River, NJ: Prentice hall, 1998"

The linear Model Predictive Control (MPC) we proposed here inherits the excellent robustness properties of the Linear Quadratic Regulators, as we discuss in Remark 12 of the Supplementary Note 6. This fact allowed us to design the control signal using a proxy (\hat{A}, \hat{B}) of the true pair (A, B) . Furthermore, in Fig. 4 of the main text, we demonstrate how the robustness of this linear MPC allows controlling large microbial communities with nonlinear population dynamics. In this analysis, we show that the linear MPC succeeds in driving the community to the desired state provided that the distance to the desired state is small (i.e., < 1 in Euclidean norm), or the interspecies interaction strengths is low (< 1), or the network connectivity is low (< 0.029), or that the proportion of driver species is larger than 0.06 (i.e., 6 of 100 species are directly controlled).

It is not possible that, globally, one can approximate a nonlinear networked dynamical system by a linear model with some fitted interaction matrix $\{\hat{A}, \hat{B}\}$. Any set of control signals calculated from the linear model will in general have no relevance to the actual system. The linear approximation is valid but only locally. For a nonlinear networked system, locally the linearized systems are different from one point to another in the phase space. The control signals from authors' MPC approach are irrelevant to the actual underlying nonlinear system.

Here we were not able to understand (or even reasonably guess) the meaning of "relevance" or "irrelevance" for a control signal, to which Reviewer #1 constantly refers. Therefore, we are not able to provide a concrete answer to this comment.

As an example, the authors stated on page 19 in the main text that the proxy interaction matrix is $\hat{A} = (-0.5, 0, -0.1; 0, -5, 1; 0, 0, 1)$ and the linearization of the population dynamics at the desired state is $A = (-0.37, 0, -0.05; 0, -5.31, 0.52; 0, 0, -1)$. The proxy interaction matrix is not even able to model the locally linear dynamics in the vicinity of the desired state (note the difference in the sign of the last element). How can one expect the proxy matrix to hold globally?

[1] Isaac Klickstein, Afroza Shirin, and Francesco Sorrentino. Locally optimal control of complex networks. *Phys. Rev. Lett.*, 119(26):268301, 2017.

As explained above, the linear MPC we proposed inherits the excellent robustness properties of the Linear Quadratic Regulators, as discussed in Remark 12 of the Supplementary Note 6. This allowed us to design the control signal using a proxy (\hat{A}, \hat{B}) of the true pair (A, B) . Furthermore, in Fig. 4 of the main text, we demonstrate how the robustness of the linear MPC allows controlling large microbial communities with nonlinear population dynamics. In Figs. 4e,f, and g of this analysis, we find that the linear MPC succeeds provided that the distance to the desired state is small (i.e., < 1 in Euclidean norm), or the interspecies interaction strength is low (< 1), or the network connectivity is low (< 0.029), or that the proportion of driver species is larger than 0.06 (i.e., 6 of 100 species are directly controlled).

Finally, although we believe that Reviewer #1 has completely misunderstood our contributions and efforts, we thank her/him again for the time reviewing our paper.

Response to Reviewer #2

Dear Authors, thank you for the great job you made during revision. My concerns have been appropriately answered. Especially Figure 4 will be great help for the Readers, I think.

We thank Reviewer #2 for her/his time reviewing our paper, and for the very insightful and constructive comments.

Response to Reviewer #3

Dear Editor,

In their extensive revisions to ms NCOMMS-18-15520A, the authors have satisfactorily addressed the concerns raised in my initial review.

We thank Reviewer #3 for her/his time reviewing our paper, and for the very constructive comments.

Response to Reviewer #4

The authors put significant effort in revising the manuscript and I am very happy on how they re-wrote the text.

We thank Reviewer #4 for her/his time reviewing our paper and for her/his very constructive comments.

There are only some minor concerns left:

1) I can see that doing experiments to validate the proposed concept are out of scope of this manuscript. It is however hard to understand that there are no data sets available that would enable a test of the findings. This expectancy might come from the title and might misguide future readers as well. There should be somewhere written that this is a theoretical framework. For example: “Inferring controllability of complex microbial communities: a network-based approach”.

We thank Reviewer #4 for this comment. Unfortunately, we cannot use the word “controllability” without introducing additional confusion, since in control theory controllability is “reserved” for the ability to drive the system between any two states (see Discussion). Thus, to emphasize that our work provides a theoretical framework to control complex microbial communities, we have revised the title of our paper as “*Controlling complex microbial communities: a theoretical framework*”. Moreover, we also revised the first paragraph of the Discussion section as follows:

Our theoretical framework allows systematically and efficiently controlling microbial communities towards desired states by identifying their driver species...

2) The authors put significant effort into explaining how they use the term ecological networks. Probably due to my background I do not think just looking into microbe-microbe interactions justify the term “ecological” here, but as highlighted in the response this was published before. I think the discussion is already out so and in the present framework given in this manuscript environmental factors could be easily implemented by future users. A sentence in the discussion on future implementations could certainly clarify this.

We thank Reviewer #4 for this comment. We have added the following sentence to the Discussion section:

...Control algorithms based on reinforcement learning [51] (RL) could provide better performance and robustness. Our characterization of minimal sets of driver species will help efficiently apply those control algorithms to microbial communities and other biological systems, as RL algorithms require specifying the “driver variables” they can actuate [51]. Here, controlling small synthetic communities could provide valuable insights for designing such controllers...

Finally, we thank Reviewer #4 again for her/his very constructive comments that have significantly improved our work.